# VibeVoice: Expressive Podcast Generation with Next-Token Diffusion

**Zhiliang Peng,**[*] **Jianwei Yu,**[*] **Wenhui Wang,**[*] **Yaoyao Chang,**[*] **Yutao Sun,**[*] **Li Dong**[*]
**Yi Zhu, Weijiang Xu, Hangbo Bao, Zehua Wang, Shaohan Huang, Yan Xia, Furu Wei**[◇]
**Microsoft Research**
https://aka.ms/GeneralAI

## Abstract

Generating long-form, multi-speaker conversational audio like podcasts poses significant challenges for traditional Text-to-Speech (TTS) systems, particularly in scalability, speaker consistency, and natural turn-taking. We present VibeVoice, a novel model designed to synthesize expressive, long-form speech with multiple speakers in a zero-shot manner. A core component of our approach is the continuous speech tokenizers operating at an ultra-low frame rate of 7.5. This tokenizer effectively preserves audio fidelity while significantly boosting computational efficiency for processing long sequences. To facilitate training on authentic conversational dynamics, we have developed an annotation pipeline that generates pseudo transcriptions and turn-taking labels for extensive podcast data. Leveraging this data and our efficient tokenizer, VibeVoice employs the next-token diffusion framework. This enables VibeVoice to: (1) synthesize long-form speech (up to 90 minutes) with up to 4 speakers, surpassing the typical 1-2 speaker limits of many prior models; and (2) achieve a high degree of naturalness in turn-taking, pacing, and the rendition of subtle non-lexical cues (such as breaths and lip smacks), which are crucial for listener immersion and capturing the authentic vibe of expressive conversations. Code and checkpoint are available at https://github.com/microsoft/VibeVoice.

## 1 Introduction

Recent advancements in Text-to-Speech (TTS) synthesis have achieved remarkable success in generating high-fidelity, natural-sounding speech for single speakers in relatively short utterances (Wang et al., 2023a; Anastassiou et al., 2024; Le et al., 2023; Chen et al., 2024a;b; Du et al., 2024; Jia et al., 2025; Ye et al., 2025; Wang et al., 2025). However, a frontier remains unexplored: the scalable synthesis of long-form, multi-speaker conversational audio, such as podcasts, multi-participant audiobooks, and extended dialogues. While commercial interest in this area is evident, with closed-source applications like Google's NotebookLM offering podcast generation capabilities, the underlying technical approaches often remain opaque. The demand for automatically generating such complex audio content from text is rapidly increasing, yet it poses substantial, open research challenges that lie beyond the capabilities demonstrated by most publicly available TTS systems.

Generating convincing multi-speaker conversations introduces unique hurdles. While traditional systems can technically produce multi-speaker, long-form audio by concatenating individually synthesized utterances (as demonstrated in our experiments), achieving naturalness in speaker interaction remains a significant challenge. Key difficulties include maintaining vocal consistency for each speaker across extended dialogues and capturing the authentic conversational flow, encompassing natural turn-taking, pacing, and subtle non-lexical cues (e.g., breaths, lip smacks) crucial for listener immersion. While previous work like MoonCast (Ju et al., 2025) has demonstrated the feasibility of synthesizing podcasts, the demand for more adaptable, efficient, and higher-fidelity architectures that can generate these interactions end-to-end persists.

To address these limitations, we present VibeVoice, as illustrated in Figure 1, a novel framework developed for the scalable synthesis of expressive, long-form, multi-speaker speech. Our approach

---

[*] Core contributors. ◇ Contact person: fuwei@microsoft.com.

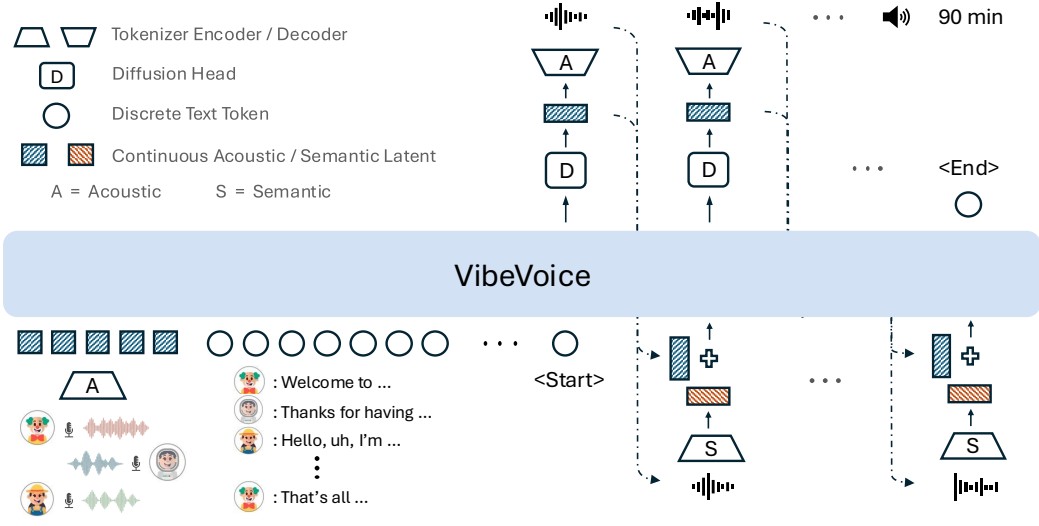

Figure 1: Scalable and expressive podcast synthesis with VIBEVOICE. Voice prompts and text scripts provide initial input. VIBEVOICE processes hybrid context features, and its hidden states condition a token level Diffusion Head (D), which predicts acoustic VAE for speech segments, subsequently recovered by acoustic decoder (A).

tackles these challenges through a synergistic combination of efficient audio representation, authentic data preparation, and an integrated generative architecture.

A cornerstone of VIBEVOICE is its efficient hybrid speech representation strategy, derived from specialized acoustic and semantic tokenizers, both operating at an ultra-low frame rate of 7.5 Hz. The Acoustic Tokenizer aggressively compresses audio while preserving remarkable reconstruction fidelity, and Semantic Tokenizer extracts linguistic content. This decoupled design, with both components leveraging the same highly efficient frame rate, allows for optimized acoustic and semantic feature extraction. These features are then combined to form the rich, yet compact, hybrid input essential for managing long-form content within our generative model. In addition, to generate long conversational speech, we propose to use hybrid speech representation in VIBEVOICE. Moreover, we developed a data processing pipeline that curates and annotates raw podcast data. This provides rich, naturalistic training material, enabling VIBEVOICE to learn realistic intonation, turn-taking, and subtle expressive cues, thereby enhancing perceived audio realism.

VIBEVOICE leverages an end-to-end LLM-based architecture with a diffusion head, drawing inspiration from LatentLM (Sun et al., 2024). The LLM (Yang et al., 2024) handles robust textual understanding and dialogue flow, while the diffusion head (Li et al., 2024b) ensures high-fidelity acoustic generation. This framework achieves scalability through efficient design, employing specialized tokenizers to create a rich yet compact hybrid input for the LLM, and a streamlined diffusion process for well-structured acoustic tokens. This architecture enables VIBEVOICE to synthesize expressive, multi-speaker podcasts with coherence, intelligibility, and engaging conversational dynamics.

## 2 METHOD

### 2.1 CONTINUOUS SPEECH TOKENIZERS

We employ two separate tokenizers as input to learn both acoustic and semantic features. It is worth noting that while we use the term 'tokenizer' to highlight its role in generating the sequence for the LLM, our Acoustic Tokenizer operates on a continuous latent space (via $\sigma$-VAE) rather than a discrete codebook. In our experiments, generating long-form speech benefits from this separate design.

**Acoustic Tokenizer** adopts the principles of a Variational Autoencoder (VAE) (Kingma & Welling, 2014), specifically drawing inspiration from the $\sigma$-VAE variant proposed in LatentLM (Sun et al., 2024) to mitigate potential variance collapse issues of VAEs when used in autoregressive modeling settings. The process involves an encoder network, parameterized by $\phi$, which maps the input audio $\boldsymbol{x}$ to the parameters of a latent distribution, primarily the mean $\mu$. Notably, variance $\sigma$ is a pre-defined distribution ($\mathcal{N}(0, C_\sigma)$) in $\sigma$-VAE, rather than a learnable distribution in VAE (Kingma & Welling, 2014), as demonstrated in Figure 2. A latent vector $\boldsymbol{z}$ is then sampled using the reparameterization trick. Following the $\sigma$-VAE approach to ensure robust variance for autoregressive modeling, we can formulate this as:

$$\mu = \text{Encoder}_\phi(\boldsymbol{x})$$
$$\boldsymbol{z} = \mu + \sigma \odot \boldsymbol{\epsilon}, \text{where } \boldsymbol{\epsilon} \sim \mathcal{N}(0, 1), \ \sigma \sim \mathcal{N}(0, C_\sigma)$$
$$\hat{\boldsymbol{x}} = \text{Decoder}_\psi(\boldsymbol{z})$$

**Semantic Tokenizer** mirrors the hierarchical architecture of the Acoustic Tokenizer's encoder, but without VAE components, as its objective is deterministic content-centric feature extraction. The main difference is the training objective, which uses Automatic Speech Recognition (ASR) as the proxy task. During training, its output is decoded by several Transformer decoder layers to predict text transcripts, aligning the semantic encoder's representations with textual semantics. Refer to Figure 2 for the comparisons.

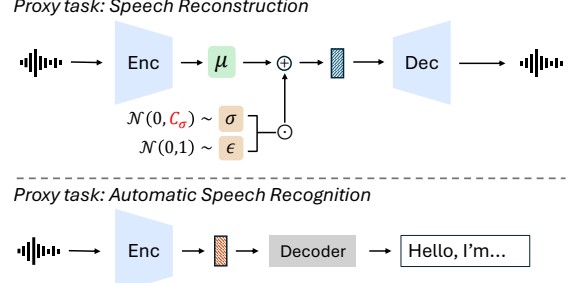

*Proxy task: Speech Reconstruction*

*Proxy task: Automatic Speech Recognition*

Figure 2: Acoustic Tokenizer (upper) reconstructs waveform by $\sigma$-VAE, while Semantic Tokenizer (lower) uses ASR as its proxy task.

## 2.2 VIBEVOICE

VIBEVOICE employs a Large Language Model (LLM) as its core sequence model, integrated with specialized audio encoding and diffusion-based decoding modules to achieve scalable, high-fidelity multi-speaker speech synthesis. The overall inference architecture is depicted in Figure 1.

### 2.2.1 INPUT REPRESENTATION

**Voice Prompt Feature** for each speaker $k \in \{1, ..., N\}$, the provided voice prompt waveform $v_k$ is encoded solely by the Acoustic Tokenizer encoder. This component, analogous to the $\sigma$-VAE described in LatentLM (Sun et al., 2024), maps the raw waveform into a sequence of continuous VAE features $\mathcal{Z}_{a,k} = \text{Acoustic}_{\text{VAE}}(v_k)$. Here, $\mathcal{Z}_{a,k} = [\boldsymbol{z}_{a,k}^1, ..., \boldsymbol{z}_{a,k}^{T_k^v}] \in \mathbb{R}^{T_k^v \times d_a}$, where $T_k^v$ is the sequence length for speaker $k$'s prompt and $d_a$ is the dimension of the acoustic features. $\mathcal{Z}_{a,k}$ capture the essential timbral and prosodic characteristics required for speaker.

**Text Script Embedding** of the dialogue script, consisting of turns $W_k$ for each speaker $k$, is tokenized into subword units. Note that script does not include the transcript of the acoustic prompt. These are then mapped to text embeddings $E_k \in \mathbb{R}^{T_k^s \times d_{llm}}$ using a standard embedding layer, where $T_k^s$ is the script length for speaker $k$ and $d_{llm}$ is the LLM's hidden dimension.

**Prompt Sequence** $X$ is formed by concatenating the acoustic-only voice prompt features and the text script embeddings, interleaved with speaker identifiers ($\text{Speaker}_k$): $X = [\text{Speaker}_1, \mathcal{Z}_{a,1}, ..., \text{Speaker}_N, \mathcal{Z}_{a,N}; \text{Speaker}_1, E_1, ..., \text{Speaker}_N, E_N, \texttt{}]$, where $\texttt{}$ is a special token of start-of-speech. $\{\mathcal{Z}_{a,k}\}_{k=1,...,N}$ are passed through a projection layer to match the LLM's hidden dimension $d_{llm}$.

### 2.2.2 SPEECH GENERATION WITH HYBRID SPEECH REPRESENTATION

The acoustic feature preserves fine-grained acoustic details, while the semantic feature captures higher-level linguistic information. To fully exploit the advantages of both, we propose to use hybrid representation in VIBEVOICE. As illustrated in Figure 1, VIBEVOICE leverages both acoustic and

semantic representations throughout the generation process. Specifically, at each generation step, the generated speech segment is predicted as

$$\boldsymbol{z}_{a,i+1} = \text{VIBEVOICE}(X, z_{p,0}, \ldots, z_{p,i}), \tag{1}$$

$$\boldsymbol{y}_{i+1} = \text{Acoustic}_{\text{Dec}}(\boldsymbol{z}_{a,i+1}), \tag{2}$$

where $\boldsymbol{y}_{i+1}$ denotes the waveform of speech segment generated at the $(i + 1)$-th step, and $\boldsymbol{z}_{p,i} \in \mathbb{R}^{d_{\text{llm}}}$ represents the hybrid speech representation. The latter is obtained by combining acoustic and semantic encodings:

$$\boldsymbol{z}_{p,i} = W_a \boldsymbol{z}_{a,i} + W_s \, \text{Semantic}_{\text{Enc}}(\boldsymbol{y}_i), \tag{3}$$

where $W_a \in \mathbb{R}^{d_{\text{llm}} \times d_a}$ and $W_s \in \mathbb{R}^{d_{\text{llm}} \times d_s}$ are learnable projection matrices. As semantic features are more close to the text prompt, involving semantic features as the input can stabilize the generation process, especially for long-form speech.

The final generated speech is obtained by concatenating the generated segment at each decoding step: $Y = \text{concat}(y_0, y_1, \ldots y_I)$. As each speech segment is generated purely based on historical context, VIBEVOICE is naturally a streaming speech generation model.

### 2.2.3 DIFFUSION-BASED ACOUSTIC LATENT VAE GENERATION

To synthesize speech, VIBEVOICE employs a light diffusion head (Li et al., 2024b) conditioned on the LLM's current hidden states. Specifically, the Diffusion Head ( $\boxed{\text{D}}$ in Figure 1) takes an LLM hidden state $\boldsymbol{h}_i$ as condition and predicts the corresponding acoustic VAE $\boldsymbol{z}_{a,i}$.

**Training:** The diffusion process (Ho et al., 2020) learns to reverse a forward process that gradually adds Gaussian noise $\epsilon \sim \mathcal{N}(0, I)$ to the clean acoustic vae $\boldsymbol{z}_{a,i}$ over $T_{\text{diff}}$ discrete timesteps $t$:

$$\boldsymbol{z}_{a,i}(t) = \sqrt{\bar{\alpha}_t} \boldsymbol{z}_{a,i} + \sqrt{1 - \bar{\alpha}_t} \epsilon \tag{4}$$

where $\bar{\alpha}_t = \prod_{s=1}^{t}(1 - \beta_s)$ is a cumulative product of terms derived from a predefined noise schedule $\{\beta_s\}_{s=1}^{T_{\text{diff}}}$. A noise prediction network $\epsilon_\theta$, constituting the lightweight Diffusion Head (Li et al., 2024b), is trained to estimate the injected noise $\epsilon$ given the noisy feature $\boldsymbol{z}_{a,i}(t)$, the timestep $t$, and the conditioning LLM hidden state $\boldsymbol{h}_i$. The objective is to minimize the L2 loss:

$$\mathcal{L}_{\text{Diff}} = \mathbb{E}_{t, \boldsymbol{z}_{a,i}, \epsilon, \boldsymbol{h}_i} ||\epsilon - \epsilon_\theta(\boldsymbol{z}_{a,i}(t), t, \boldsymbol{h}_i)||^2 \tag{5}$$

Crucially, the diffusion process is trained to predict only the acoustic VAE $\boldsymbol{z}_{a,i}$. Concurrently, when the LLM generates the conditioning hidden state $\boldsymbol{h}_i$ for the diffusion head, it is also tasked with predicting whether the current speech segment should conclude, effectively determining if a diffusion termination token (e.g., <E>) should be emitted at this step.

**Inference:** During synthesis at step $i$, the Diffusion Head utilizes Classifier-Free Guidance (CFG) (Ho & Salimans, 2022) to enhance conditioning. The process starts with random noise and iteratively denoises it. At each denoising step $t$, two predictions of the noise are made by the network $\epsilon_\theta$: (1) The conditional prediction, using the current predicted token's hidden state $\boldsymbol{h}_i$; and (2) The unconditional prediction, where represents a context derived solely from a special start-of-sequence token , effectively an unconditional or minimally conditioned generation prompt. The final noise estimate used for the denoising update is a linear combination of these two:

$$\hat{\epsilon} = \epsilon_\theta(\boldsymbol{z}_{a,i}(t), t, \boldsymbol{h}_{}) + w \cdot (\epsilon_\theta(\boldsymbol{z}_{a,i}(t), t, \boldsymbol{h}_i) - \epsilon_\theta(\boldsymbol{z}_{a,i}(t), t, \boldsymbol{h}_{})) \tag{6}$$

where $w$ is the guidance scale. $w = 0$ recovers purely unconditional generation (based on ), while $w > 1$ amplifies the conditioning from. This guided noise estimate $\hat{\theta}$ is then used to take a step towards the less noisy feature. This iterative process, typically accelerated using an efficient sampler like DPM-Solver++ (Lu et al., 2025), eventually yields an estimate of clean acoustic features.

### 2.3 DATA PREPARATION

The development of the proposed VIBEVOICE relies on speech data with consistent long-range annotations, including transcriptions and speaker turn labels. However, previous methods have primarily focused on generating short audio segments (Yu et al., 2024; He et al., 2024), which

are not directly applicable to multi-speaker, long-form audio annotation. To address this limitation, we propose a novel automatic annotation pipeline tailored for extended speech data. The pipeline consists three steps, including **Segmentation and Transcription**, **Diarization**, **Quality Filtering**. Detials and evaluations are available in Appendix A.

Note that, compared to previous data processing pipelines (Yu et al., 2024; He et al., 2024; Ding et al., 2025), our pipeline does not include speech enhancement. This decision is based on our observation that while speech enhancement can effectively reduce noise, it often introduces distortion to the speech signal. In particular, emotionally expressive cues such as interjections or prosodic elements are prone to degradation, which negatively impacts the naturalness of the audio.

## 3 EXPERIMENT

### 3.1 TOKENIZER SETUP

**Acoustic Tokenizer** is architected with a mirror-symmetric encoder-decoder structure. The encoder employs a hierarchical design with 7 stages of modified Transformer blocks (Vaswani et al., 2017) (using 1D depth-wise causal convolutions instead of self-attention) for efficient streaming processing. Six downsampling layers achieve a cumulative 3200x downsampling rate from a 24kHz input, yielding 7.5 tokens/frames per second. Each encoder/decoder component has approximately 340M parameters. The training objective follows the DAC methodology (Kumar et al., 2023), including its discriminator and loss designs. Key hyperparameters and further training specifics are detailed in Appendix F.

**Semantic Tokenizer**'s encoder mirrors the architecture of the Acoustic Tokenizer for structural consistency. It is trained exclusively on an ASR proxy task using a cross-entropy loss function. This objective compels the encoder to capture explicitly semantic and phonetic information. Upon completion of pre-training, the decoder—which serves only to facilitate this training process—is discarded. The keys hyperparameters are the same as Acoustic Tokenizer.

### 3.2 VIBEVOICE SETUP

We instantiated VIBEVOICE's core Large Language Model (LLM) using 1.5B and 7B parameter versions of Qwen2.5 (Yang et al., 2024). The diffusion head (Li et al., 2024b) comprises 4 layers (approx. 123M parameters for 1.5B version). During VIBEVOICE training, the pre-trained acoustic and semantic tokenizers remained frozen, with only the LLM and diffusion head parameters being learnable. We employed a curriculum learning strategy for the LLM input sequence length, progressively increasing from 4,096 to 65,536 tokens over 110k training steps (4,096 tokens for the first 40k steps, then 16,384 tokens for steps 40k-80k, then 32,768 tokens for steps 80k-100k, and finally 65,536 tokens for steps 100k-110k). The 7B version discarded the final phase due to resource limitations. VIBEVOICE was trained on approximately 80 billion tokens from an internal pseudo-labeled podcast audio collection (Section 2.3). Training a 1.5B model took approximately 170 hours on 64 AMD Instinct MI300X GPUs by using nnscaler training engine (Lin et al., 2024). Comprehensive training hyperparameters are listed in Appendix F. During inference, the CFG scale and denoising step are set to 1.3 and 10 respectively. Inference time comparison are at Appendix E.

### 3.3 EVALUATION SETUP

#### 3.3.1 VIBEVOICE EVALUATION

**Objective Evaluation:** Recognizing the limitations of existing public benchmarks for the podcast generation task (e.g., the MoonCast evaluation set (Ju et al., 2025), which contains only four samples), we developed VIBEVOICE-Eval. This dataset was curated using our data processing pipeline (described in Section 2.3 and Appendix A) to better capture diverse podcast scenarios, and it consists of 108 podcast samples with durations ranging from 1 to 30 minutes. Details can be found in Appendix G.

We employed Word Error Rate (WER) and a widely used speaker similarity metric (SIM-O) (Anastassiou et al., 2024; Ju et al., 2025) as objective metrics to assess the quality of podcast generation. All generated podcasts were annotated using our custom data pipeline. The only difference is that we

| Model | Subjective | | | | Objective | | |
|---|---|---|---|---|---|---|---|
| | **Realism** | **Richness** | **Preference** | **Average** | **WER-W** | **WER-N** | **SIM-O** |
| Cosyvoice2 (Du et al., 2024) | - | - | - | - | 3.45 | 3.86 | 0.68 |
| Mooncast (Ju et al., 2025) | - | - | - | - | 2.81 | 3.29 | 0.562 |
| SesameAILabs-CSM (SesameAILabs, 2025) | 2.89 $\pm1.15$ | 3.03 $\pm1.11$ | 2.75 $\pm1.08$ | 2.89 $\pm1.12$ | 2.66 | 3.05 | 0.685 |
| Higgs Audio V2 (Boson AI, 2025) | 2.95 $\pm1.13$ | 3.19 $\pm1.06$ | 2.83 $\pm1.16$ | 2.99 $\pm1.13$ | 5.94 | 5.97 | 0.543 |
| Elevenlabs v3 alpha (Elevenlabs) | 3.34 $\pm1.11$ | 3.48 $\pm1.05$ | 3.38 $\pm1.12$ | 3.40 $\pm1.09$ | 2.39 | 2.47 | 0.623 |
| Gemini 2.5 pro preview tts (Google) | 3.55 $\pm1.20$ | 3.78 $\pm1.11$ | 3.65 $\pm1.15$ | 3.66 $\pm1.16$ | 1.73 | 2.43 | - |
| VIBEVOICE-1.5B | 3.59 $\pm0.95$ | 3.59 $\pm1.01$ | 3.44 $\pm0.92$ | 3.54 $\pm0.96$ | **1.11** | **1.82** | 0.548 |
| VIBEVOICE-7B | **3.71** $\pm0.98$ | **3.81** $\pm0.87$ | **3.75** $\pm0.94$ | **3.76** $\pm0.93$ | 1.29 | 1.95 | **0.692** |

Table 1: Human subjective and objective evaluation results. WER-W means using Whisper while WER-N means using Nemo. For all subjective metrics and SIM-O, higher scores are better. For WER, lower scores are better. Best results are in **bold**. The first phase subjective evaluation of Cosyvoice2, Mooncast and VIBEVOICE-1.5B can be found in Appendix I.

use the ground-truth number of speakers as the number of cluster centers in the diarization process. Since the diarization results may not align perfectly with the input text prompts, we assign speaker labels based on which cluster center has the highest similarity to the speaker embedding of each speech prompt, and then compute SIM-O accordingly.

**Subjective Evaluation:** In the subjective evaluation, we used eight podcast samples. These samples were based on two-speaker dialogue texts generated by a large language model, covering diverse topics such as technology, art, and travel. The lengths of the samples ranged from 3 to 15 minutes, with an average duration of approximately 7.3 minutes.

For subjective evaluation, we recruited 24 human annotators to provide Mean Opinion Scores (MOS) across three dimensions: **Realism** (how natural and human-like the speech sounds, including prosody, emotion, and the smoothness of speaker turns), **Richness** (the expressiveness of the speech in terms of tone and emotion, including variation and adaptation to context), and **Preference** (overall listener enjoyment and subjective preference, reflecting naturalness, pleasantness, and engagement). The evaluation covered six models with all eight test samples, meaning that each annotator listened to approximately **six hours of audio** in total (roughly equivalent to reviewing 2,160 short utterances of 10 seconds). Detailed definitions for each dimension and the precise 5-point scoring rubrics provided to evaluators are available in Appendix H.

### 3.3.2 TOKENIZER EVALUATION

The fidelity of audio reconstructed from acoustic tokens is a critical indicator of the tokenizer's efficacy in preserving essential acoustic information, particularly under high compression rates. To quantify this, we employed a suite of widely recognized objective metrics: PESQ (Rix et al., 2001) to measure overall speech quality, STOI (Taal et al., 2010) to assess speech intelligibility, and UTMOS (Saeki et al., 2022), which yields scores highly correlated with human evaluations. These evaluations were conducted on both the LibriTTS test-clean and test-other datasets (Zen et al., 2019), and LibriSpeech test-clean dataset (Panayotov et al., 2015).

## 4 RESULTS

### 4.1 PODCAST GENERATION COMPARISON

We report subjective evaluation results in Table 1 and objective evaluation on the VIBEVOICE-Eval dataset in Table 2, comparing VIBEVOICE with baseline models and ablating different tokenizer configurations and model sizes.

**Comparison with Top-tier Models:** In human-led subjective assessments, VIBEVOICE-7B attains the highest average score (3.76). It consistently outperformed all other models across the dimensions of Realism (3.71), Richness (3.81), and listener Preference (3.75). Notably, it surpassed strong proprietary models such as Google's Gemini 2.5 Pro preview TTS (3.66 average) and Elevenlabs v3 alpha (3.40 average), indicating its superior ability to generate natural and engaging conversational audio that is preferred by human listeners. From an objective standpoint, VIBEVOICE also excels.

| Model | Seq. Leng. | 1 Speaker | | 2 Speakers | | 3 Speakers | | 4 Speakers | | Overall | |
|---|---|---|---|---|---|---|---|---|---|---|---|
| | | WER-W↓ | SIM-O↑ | WER-W↓ | SIM-O↑ | WER-W↓ | SIM-O↑ | WER-W↓ | SIM-O↑ | WER-W↓ | SIM-O↑ |
| VIBEVOICE-Eval Short (0∼12 min) Set | | | | | | | | | | | |
| Cosyvoice2 - Concat | | 3.14 | **0.79** | 3.5 | 0.73 | 5.33 | 0.69 | 5.83 | 0.70 | 4.27 | 0.73 |
| MoonCast | 40K | 7.2 | 0.61 | 7.9 | 0.63 | 17.2‡ | ‡ | 11.5‡ | 0.48‡ | 10.4‡ | 0.55‡ |
| **Tokenizer Ablation** | | | | | | | | | | | |
| Acoustic (1.5B) | 16K | 1.06 | 0.7 | 6.15 | 0.7 | 13.74 | 0.70 | 6.46 | 0.64 | 6.22 | 0.68 |
| Hybrid (1.5B) | 16K | 1.93 | 0.66 | 0.79 | 0.59 | 2.50 | 0.64 | 1.68 | 0.64 | 1.84 | 0.64 |
| **VIBEVOICE-1.5B** | 32K | 1.32 | 0.60 | 0.91 | 0.60 | 4.86 | 0.56 | 1.46 | 0.59 | 2.11 | 0.59 |
| **VIBEVOICE-1.5B** | 64K | 0.63 | 0.63 | 1.92 | 0.59 | 1.48 | 0.58 | 1.34 | 0.58 | 1.22 | 0.60 |
| **VIBEVOICE-7B** | 32K | **0.47** | 0.76 | **0.53** | 0.75 | **0.68** | 0.75 | **1.02** | **0.72** | **0.66** | **0.75** |
| VIBEVOICE-Eval Long (12∼30 min) Set | | | | | | | | | | | |
| Cosyvoice2 - Concat | | 5.76 | **0.80** | 4.94 | 0.75 | 4.34 | 0.71 | 4.77 | 0.70 | 4.95 | 0.74 |
| MoonCast | 40K | - | - | 13.64* | 0.67* | - | - | - | - | - | - |
| **VIBEVOICE-1.5B** | 32K | **0.87** | 0.57 | 2.08 | 0.61 | 1.06 | 0.57 | **1.33** | 0.54 | 1.33 | 0.57 |
| **VIBEVOICE-1.5B** | 64K | 1.80 | 0.63 | 1.59 | 0.62 | 0.97 | 0.60 | 1.80 | 0.56 | 1.55 | 0.59 |
| **VIBEVOICE-7B** | 32K | 1.08 | 0.79 | **1.55** | **0.77** | **0.84** | **0.73** | 1.51 | **0.71** | **1.24** | **0.75** |

Table 2: Results of VIBEVOICE and baseline models on the VIBEVOICE-Eval dataset for multi-speaker podcast generation. Results are presented for short (0∼12 min) and long (12∼30 min) duration subsets, across varying speaker counts. Seq. Leng. denotes the LLM training sequence length. "∗" denotes using a subset (12–13 min), and "‡" denotes using the successful cases only (with 3 retries) due to MoonCast crashes on long and multi-speaker(≥ 3) generation.

While the 1.5B model delivered the lowest Word Error Rate (WER-W: 1.11), demonstrating superior intelligibility, the VIBEVOICE-7B model achieved the highest speaker similarity score (SIM-O: 0.692) among all evaluated systems. This highlights its exceptional capability to preserve the unique vocal characteristics of a target speaker while maintaining high content accuracy.

**Scalability in Long-Form and Multi-Speaker Synthesis:** A key contribution of VIBEVOICE is its capacity to generate coherent, long-form audio with multiple speakers, a significant challenge for existing TTS systems. As shown in Table 2, VIBEVOICE demonstrates remarkable stability and consistency in extended scenarios. For long-duration podcasts (12–30 minutes), VIBEVOICE-7B maintains a very low WER-W of 1.24 and a high SIM-O of 0.75. This contrasts sharply with baseline models; for instance, MoonCast was unable to process the full long-form test set and frequently crashed, as noted in the table. Furthermore, based on supplementary analysis, VIBEVOICE is architected to support up to four distinct speakers for durations of up to 30 minutes. This capability substantially exceeds that of prior work like MoonCast, which is limited to two speakers and up to 10 minutes of audio. This enhanced scalability is a direct result of our novel next-token diffusion framework and low frame rate continuous speech tokenizer, which operates at 7.5 Hz.

## 4.2 ABLATION STUDIES

**Tokenizer Configurations:** The choice to employ a hybrid tokenizer that fuses separate acoustic and semantic representations is critical to model performance. The "Tokenizer Ablation" results in Table 2 reveal that an "Acoustic-only" model, while achieving competitive speaker similarity (SIM-O: 0.68), suffers from a significantly degraded multi-speaker WER (6.22 overall). This indicates that while acoustic features alone can preserve speaker timbre, they lack the necessary semantic guidance to maintain content coherence in interactive dialogues. The final "Hybrid" approach consistently strikes a superior balance, dramatically improving intelligibility (WER: 1.84) while maintaining strong speaker identity (SIM-O: 0.64).

**Model Scale:** Scaling the model from 1.5B to 7B parameters yields substantial and comprehensive performance gains. This is demonstrated by a clear listener preference for the larger model, with the average subjective score (Table 1) increasing from 3.54 to 3.76. This perceptual enhancement is mirrored by dramatic improvements in objective metrics (Table 2), where the overall WER-W dropped from 2.11 to 0.66 and the SIM-O score rose from 0.59 to 0.75. These results confirm that a larger model significantly boosts both perceived audio quality and technical accuracy in speaker preservation.

**CFG and DDPM Steps:** Figure 3 illustrates the ablation of Classifier-Free Guidance (CFG) scale and DDPM (Ho et al., 2020) denoising steps on inference performance. For WER (Figure 3a), optimal results (e.g., WER of 1.55) are achieved with 10 denoising steps and a CFG scale of 1.25. Using fewer steps (e.g., 5) significantly degrades WER, while excessive steps show diminishing

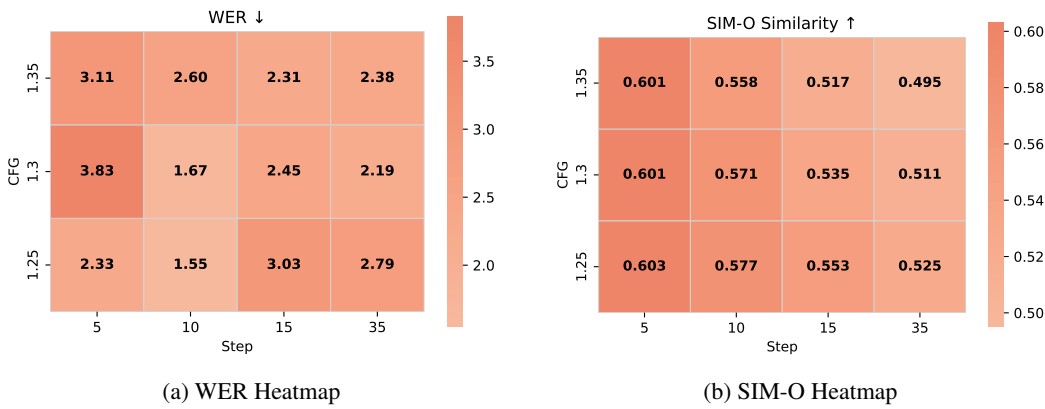

(a) WER Heatmap          (b) SIM-O Heatmap

Figure 3: Ablation of CFG and DDPM steps on WER and SIM-O. Heatmaps show the effect of DDPM steps (x-axis) and classifier-free guidance (CFG) scale (y-axis) on SIM-O and WER scores.

| Model | $N_q$ | Token Rate | Test-clean | | | Test-other | | |
|---|---|---|---|---|---|---|---|---|
| | | | PESQ | STOI | UTMOS | PESQ | STOI | UTMOS |
| Ground-Truth | - | - | - | - | 4.056 | - | - | 3.483 |
| Encodec (Défossez et al., 2022) | 8 | 600 | 2.72 | **0.939** | 3.04 | 2.682 | **0.924** | 2.657 |
| DAC (Kumar et al., 2023) | 4 | 400 | 2.738 | 0.928 | 3.433 | 2.595 | 0.908 | 2.945 |
| Encodec (Défossez et al., 2022) | 4 | 300 | 2.052 | 0.901 | 2.307 | 2.052 | 0.884 | 2.088 |
| SpeechTokenizer (Zhang et al., 2023) | 4 | 300 | 1.931 | 0.878 | 3.563 | 1.737 | 0.837 | 3.018 |
| DAC (Kumar et al., 2023) | 1 | 100 | 1.246 | 0.771 | 1.494 | 1.245 | 0.751 | 1.499 |
| WavTokenizer (Ji et al., 2025) | 1 | 75 | 2.373 | 0.914 | 4.049 | 2.261 | 0.891 | 3.431 |
| WavTokenizer (Ji et al., 2025) | 1 | 40 | 1.703 | 0.862 | 3.602 | 1.662 | 0.834 | 3.055 |
| Ours (Acoustic) | N/A | 7.5 | **3.068** | 0.828 | **4.181** | **2.848** | 0.823 | **3.724** |

Table 3: Objective evaluation of reconstruction quality on the LibriTTS test-clean and test-other datasets. $N_q$ denotes the number of quantizers; VIBEVOICE uses a single continuous $\sigma$-VAE. Token Rate indicates the number of tokens/frames generated per second of audio. Higher PESQ, STOI, and UTMOS scores indicate better performance. Best results are in **bold**.

returns. For SIM-O Similarity (Figure 3b), the highest scores (around 0.6) are obtained with as few as 5 DDPM steps across all tested CFG scales, with a slight decrease observed as steps increases. Additional visualizations are provided in Appendix J.

## 4.3 RECONSTRUCTION COMPARISON

Table 3 details the objective acoustic reconstruction performance of our tokenizer against existing methods (Défossez et al., 2022; Kumar et al., 2023; Zhang et al., 2023; Ji et al., 2025) on the LibriTTS dataset (Zen et al., 2019). Results on LibriSpeech (Panayotov et al., 2015) can be found in Appendix D. Our acoustic tokenizer, uniquely operating at an ultra-low 7.5 Hz, achieves leading PESQ and UTMOS scores on both test-clean (PESQ: 3.068, UTMOS: 4.181) and test-other (PESQ: 2.848, UTMOS: 3.724) subsets. This demonstrates its capacity for high-fidelity, perceptually excellent audio reconstruction despite aggressive compression, which is a key factor for VIBEVOICE's scalability with long-form audio. Regarding acoustic tokenizer, while its PESQ and UTMOS scores are strong, its STOI scores (test-clean: 0.828, test-other: 0.823) are comparatively lower. This is likely due to our primary training data (podcasts) not undergoing extensive noise reduction, which can impact STOI's intelligibility focus more than PESQ/UTMOS's assessment of overall perceived quality.

## 5 RELATED WORK

**Zero-shot Text-to-Speech (TTS)** aims to synthesize speech in a target speaker's voice using only a brief voice prompt from that speaker, without requiring speaker-specific model fine-tuning. Existing approaches for single-speaker zero-shot TTS can be broadly categorized into multi-stage (Wang

et al., 2023a; Anastassiou et al., 2024) and one-stage methods (Le et al., 2023; Chen et al., 2024a;b; Jia et al., 2025; Ye et al., 2025; Wang et al., 2025; Shen et al., 2023; Ju et al., 2024; Borsos et al., 2023; Zhu et al., 2024). Multi-stage models typically first predict coarse-grained representations, such as semantic tokens (Wang et al., 2023a; Anastassiou et al., 2024), and subsequently employ a non-autoregressive model (Wang et al., 2023a) or a generative diffusion model (Anastassiou et al., 2024) for coarse-to-fine acoustic feature generation. In contrast, one-stage methods simplify this pipeline by directly predicting discrete codes (Wang et al., 2025; Ye et al., 2025), Mel spectrograms (Meng et al., 2024), or continuous VAE latent features (Jia et al., 2025).

While these methods have achieved considerable success for individual utterances, extending zero-shot capabilities to complex, interactive scenarios like multi-party dialogues and podcasts presents significant further challenges. Accurately modeling natural multi-speaker dynamics—including seamless turn-taking, consistent inter-speaker prosody, and subtle paralinguistic cues—is paramount (Schuller et al., 2013; Zhang et al., 2019; Nguyen et al., 2023; Mitsui et al., 2023; Zhang et al., 2024). Furthermore, capturing the spontaneity inherent in natural human conversations, such as filler words, hesitations, and informal speech patterns, remains important aspect for high-fidelity synthesis (Li et al., 2024a;c).

**Podcast Generation** inherently grapples with a range of complex challenges, including preserving distinct speaker identities and emotional states over long durations, ensuring natural transitions and interactions among multiple participants, and maintaining a sense of spontaneity in dialogue. A pioneering effort in this direction is NotebookLM[1], though it lacks an accompanying academic paper or technical report and supports only two fixed speaker voice. Another related work, SpeechSSM (Park et al., 2024), can generate audio up to 16 minutes in length but is limited to single-speaker scenarios. While recent advances such as MoonCast (Ju et al., 2025), FireredTTS2 (Xie et al., 2025), Higgs Audio V2 (Boson AI, 2025), Moss-TTSD (OpenMOSS Team, 2025) and DSM-TTS (Zeghidour et al., 2025) have made notable progress in addressing these multifaceted challenges, there remains a pressing need for more scalable and flexible solutions to podcast generation, such as generation length, number of speakers and generation stability.

**Speech Tokenizer** has become a cornerstone in modern speech processing, enabling efficient representation of audio for diverse applications. Early works with discrete representations, pioneered by VQ-VAE (van den Oord et al., 2017) and significantly advanced by subsequent neural audio codecs like SoundStream (Zeghidour et al., 2021) and Encodec (Défossez et al., 2022), demonstrated the feasibility of high-quality speech reconstruction with low-bit rate discrete tokens. Then, significant research efforts have been dedicated to improving reconstruction fidelity through advancements in discriminator architectures (Wu et al., 2023; Kumar et al., 2023; Ahn et al., 2024), alongside efforts to further improve compression ratios (Ji et al., 2025; Shechtman & Dekel, 2024) and frame rate (Zeng et al., 2024). Increasingly, there is a trend towards enhancing the semantic properties of the tokenizer's latent space to better support generative tasks. Some approaches aim to enrich semantic content within a unified encoder structure (Zhang et al., 2023; Défossez et al., 2024), while others explore separate encoders (Wang et al., 2025; Ye et al., 2025).

Our tokenizer builds upon these directions with two key design choices: firstly, it operates at an ultra-low frame rate of 7.5 Hz to achieve substantial compression and efficiency; secondly, it explicitly decouples the acoustic and semantic tokenization pathways, employing distinct tokenizers that specialize in their respective domains to ensure both high-fidelity audio reconstruction and rich semantic grounding for the synthesis process.

## 6 CONCLUSION

We introduced VIBEVOICE, a novel framework for zero-shot, expressive, long-form, multi-speaker podcast generation. By integrating efficient hybrid speech representations from specialized ultra-low frame rate (7.5 Hz) acoustic and semantic tokenizers with an end-to-end LLM-based next-token diffusion architecture, and leveraging a curated podcast dataset, VIBEVOICE achieves state-of-the-art performance. It scalably synthesizes high-quality audio for up to 90 minutes with up to 4 speakers, demonstrably surpassing existing baselines in both subjective perceptual

---

[1]https://notebooklm.google/

quality—including naturalness, coherence, and realism—and objective metrics like WER, thereby significantly advancing the capabilities of conversational TTS.

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

## A  DATA PREPARATION PIPELINE

The data preparation pipeline contains the following steps:

**Segmentation And Transcription** First, we use the silero voice activity detection (VAD) tool to segment long audio recordings into shorter clips, each with a maximum duration of 30 seconds. Next, we apply Whisper-large-v3-turbo (Radford et al., 2023) to generate transcriptions with punctuation, along with word-level timestamps. Since punctuation marks (specifically [.?!]) are closely related to speaker turns, we further perform semantic-based re-segmentation by splitting the original audio at their end timestamps. This approach results in more accurate segmentation of speaker boundaries.

**Diarization** We use the vblinkp model from the WeSpeaker toolkit (Wang et al., 2023b) to perform speech diarization. Specifically, the audio segments obtained from the previous step are first divided into frames using a window length of 1.5 seconds and a hop size of 0.75 seconds. Speaker embeddings are then extracted for each frame, and HDBSCAN (Campello et al., 2013) clustering is applied to assign speaker labels. To enhance clustering consistency, we further merge clusters whose centroids have a cosine similarity greater than 0.67. Finally, we re-segment the audio based on the frame-level speaker labels, yielding the final speaker turn annotations.

**Quality Filtering** To further ensure the accuracy of the annotations, we apply the following post-filtering strategies. First, we employ a secondary ASR model (Xu et al., 2023) to re-transcribe the final segments. If the word error rate (WER) between the primary and secondary ASR outputs exceeds 20%, the corresponding segment is considered unreliable. If more than 30% of the segments in a long audio file are marked as unreliable, the entire audio is discarded. Second, we filter out audio files in which the total duration of speech is less than 60% of the overall audio length. Third, we discard audio files that contain more than 4 distinct speakers.

Table 4 summarizes the open-source toolkits utilized in our data preparation (Section 2.3) and evaluation (Section 3.3.1). We gratefully acknowledge all contributors to these open-source projects for their valuable and generous work.

## B  MORE TOKENIZER ABLATIONS

To validate the necessity of our hybrid representation, we analyze the Coupled Tokenizer, an alternative design explored during development. This architecture allows us to investigate the effects of encoding semantic and acoustic features within a single, shared latent space, providing a direct contrast to our final disentangled hybrid approach.

As shown in Figure 4, the Coupled Tokenizer employs a single encoder to map input waveforms into a shared latent distribution $\mu$. This unified representation serves dual objectives: (1) speech reconstruction via an Acoustic Decoder, and (2) transcript prediction via a Semantic Decoder. This joint optimization aims to capture both acoustic fidelity and semantic content within a unified bottleneck.

| Model | URL Link |
|---|---|
| | **Data Processing Pipeline** |
| Silero VAD | `https://github.com/snakers4/silero-vad` |
| Whisper-large-v3-turbo | `https://huggingface.co/openai/whisper-large-v3-turbo` |
| Nemo ASR | `https://huggingface.co/nvidia/parakeet-tdt-1.1b` |
| Wespeaker embedding | `https://github.com/wenet-e2e/wespeaker` |
| | `https://wenet.org.cn/downloads?models=wespeaker&version=voxblink2_samresnet100.zip` |
| | **Evaluation** |
| WER toolkit | `https://github.com/QwenLM/Qwen-Audio/blob/main/eval_audio/evaluate_asr.py#L101` |
| SIM-O toolkit | `https://github.com/BytedanceSpeech/seed-tts-eval/blob/main/cal_sim.sh` |

Table 4: List of models and tools used in our data processing pipeline.

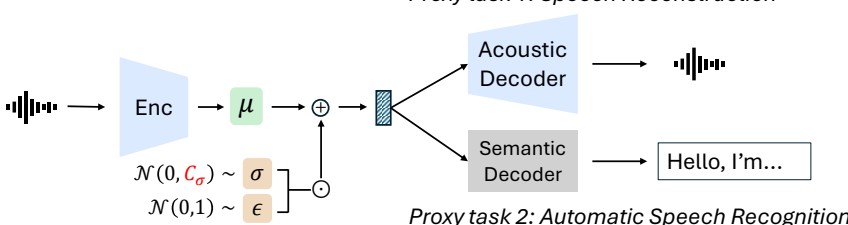

Figure 4: An illustration of the Coupled Tokenizer architecture. A single encoder produces a shared latent representation $\mu$, which is utilized for both speech reconstruction (Acoustic Decoder) and ASR (Semantic Decoder). This design contrasts with our final Hybrid architecture (Figure 2), which employs separate encoders to decouple semantic and acoustic representations.

We evaluated this design on both reconstruction quality and downstream podcast generation performance. Our findings are as follows:

1. **Reconstruction Quality:** As shown in Table 7, the reconstruction performance of the Coupled Tokenizer (STOI: 0.72, PESQ: 1.92) is substantially lower than that of the acoustic encoder from our Hybrid model (STOI: 0.75, PESQ: 2.98). This suggests that, at low framerates such as 7.5 Hz, forcing a single latent space to encode both abstract semantics and acoustic information fundamentally undermines its ability to retain the fine-grained acoustic details needed for high-fidelity reconstruction.

2. **Podcast Generation Performance:** The results of integrating the Coupled Tokenizer into our generation framework are detailed in Table 5. While the Coupled model improves content stability over the Acoustic-only baseline (reducing overall WER from 6.22 to 3.55), this improvement comes at a severe cost to speaker identity. The model yields an overall speaker similarity (SIM-O) of just 0.45—the lowest of all settings. This is a direct consequence of its degraded acoustic modeling capability; as evidenced by its poor reconstruction scores (Table 7), the shared latent space fails to preserve the unique acoustic details of a speaker's voice.

These results clarify the architectural trade-offs summarized in Table 5. The **Acoustic-only** baseline relies on entangled representations; while efficient for single speakers, it fails to disentangle speaker identity from content in multi-speaker scenarios, leading to semantic collapse (WER spikes from 1.06 to 13.74). The **Coupled** model introduces semantic constraints to mitigate this but over-compresses the representation, sacrificing speaker fidelity for content stability. Consequently, our final **Hybrid** architecture proves necessary: by explicitly disentangling semantic structure from acoustic realization, it achieves the optimal balance, maintaining both high intelligibility and robust speaker identity across complex multi-speaker sessions.

| Tokenizer | Seq. Leng. | 1 Speaker | | 2 Speakers | | 3 Speakers | | 4 Speakers | | Overall | |
|---|---|---|---|---|---|---|---|---|---|---|---|
| | | WER-W↓ | SIM-O↑ | WER-W↓ | SIM-O↑ | WER-W↓ | SIM-O↑ | WER-W↓ | SIM-O↑ | WER-W↓ | SIM-O↑ |
| Acoustic | 16K | 1.06 | 0.7 | 6.15 | 0.7 | 13.74 | 0.70 | 6.46 | 0.64 | 6.22 | 0.68 |
| Coupled | 16K | 6.79 | 0.43 | 1.18 | 0.47 | 1.4 | 0.48 | 2.6 | 0.45 | 3.55 | 0.45 |
| Hybrid (Final) | 16K | 1.93 | 0.66 | 0.79 | 0.59 | 2.50 | 0.64 | 1.68 | 0.64 | 1.84 | 0.64 |

Table 5: Ablation study on tokenizer architectures for podcast generation (no more than 12 minutes). The final Hybrid approach is evaluated against Acoustic-only and Coupled baselines. The results show that our Hybrid approach achieves the best overall balance of WER and higher SIM-O, especially in multi-speaker scenarios.

## C  MORE BENCHMARK RESULTS

We evaluate VIBEVOICE on the SEED test sets (Anastassiou et al., 2024), a widely used benchmark composed of short utterances. For evaluation, approximately 1,000 English samples and 2,000 Chinese samples are drawn from the CommonVoice dataset, denoted as *test-en* and *test-zh*, respectively. We compute word error rate (WER) using Whisper-large-v3 for *test-en* and Paraformer (Gao et al., 2022) for *test-zh*. For speaker similarity (SIM), we adopt a WavLM-large (Chen et al., 2022) model.

Table 6 presents the results on the SEED test sets. Although our model is primarily trained on long-form speech, it demonstrates strong generalization on short-utterance benchmarks. In addition, by employing a lower frame rate, our model substantially reduces the number of decoding steps required to synthesize one second of speech.

| Model | Frame Rate | test-zh | | test-en | |
|---|---|---|---|---|---|
| | | CER(%)↓ | SIM↑ | WER(%)↓ | SIM↑ |
| MaskGCT (Wang et al., 2024) | 50 | 2.27 | 0.774 | 2.62 | 0.714 |
| Seed-TTS (Anastassiou et al., 2024) | - | 1.12 | 0.796 | 2.25 | 0.762 |
| FireRedTTS (Guo et al., 2024) | 25 | 1.51 | 0.635 | 3.82 | 0.460 |
| CosyVoice 2 (Du et al., 2024) | 25 | 1.45 | 0.748 | 2.57 | 0.652 |
| Spark TTS (Wang et al., 2025) | 50 | 1.20 | 0.672 | 1.98 | 0.584 |
| VIBEVOICE-1.5B | 7.5 | 1.16 | 0.744 | 3.04 | 0.689 |

Table 6: Results on the SEED test sets.

## D  MORE RECONSTRUCTION RESULTS

The acoustic reconstruction performance on the LibriSpeech test-clean set (Table 7) largely mirrors and reinforces the conclusions drawn from our LibriTTS evaluations (Table 3).

Our dedicated VIBEVOICE Acoustic Tokenizer, operating at an exceptionally low 7.5 fps with a single quantizer ($N_q = 1$), achieves a UTMOS score of 4.19. This is highly competitive, even surpassing models like BigCodec (Xin et al., 2024) (UTMOS 4.11 at 80 fps) and achieving similar perceptual quality to X-codec2 (Ye et al., 2025) (UTMOS 4.13 at 50 fps) and BiCodec (Wang et al., 2025) (UTMOS 4.18 at 50 fps), despite those operating at nearly 7 times our frame rate. This underscores its remarkable efficiency in delivering high perceptual quality with significantly fewer tokens per second, a trend consistent with its leading PESQ and UTMOS performance on LibriTTS.

Conversely, the Coupled tokenizer (demonstrated in Figure 4) demonstrates a notable degradation in reconstruction quality (UTMOS 3.3 on LibriSpeech vs. 4.19 for the acoustic variant). This performance, while still reasonable for such an extreme frame rate, is significantly lower than the dedicated acoustic model. These findings further solidify our rationale for employing separate acoustic and semantic tokenizers in the final VIBEVOICE architecture to preserve acoustic fidelity.

## E  VIBEVOICE INFERENCE TIME

To understand the computational cost associated with generating speech using VIBEVOICE, we profiled the inference time of its core components on a single NVIDIA A6000 GPU. Table 8

| Model | Frame Rate | $N_q$ | STOI | PESQ | UTMOS |
|---|---|---|---|---|---|
| Ground Truth | – | – | – | – | 4.09 |
| DAC (Kumar et al., 2023) | 600 | 12 | 0.95 | 4.01 | 4.00 |
| Encodec (Défossez et al., 2022) | 600 | 8 | 0.94 | 2.77 | 3.09 |
| Encodec (Défossez et al., 2022) | 150 | 2 | 0.85 | 1.56 | 1.58 |
| DAC (Kumar et al., 2023) | 100 | 2 | 0.73 | 1.13 | 1.29 |
| SpeechTokenizer (Zhang et al., 2023) | 100 | 2 | 0.77 | 1.25 | 2.28 |
| Mimi (Défossez et al., 2024) | 100 | 8 | 0.91 | 2.25 | 3.56 |
| BigCodec (Xin et al., 2024) | 80 | 1 | 0.93 | 2.68 | 4.11 |
| WavTokenizer (Ji et al., 2025) | 75 | 1 | 0.90 | 2.13 | 3.79 |
| Mimi (Défossez et al., 2024) | 75 | 6 | 0.89 | 1.99 | 3.38 |
| SpeechTokenizer (Zhang et al., 2023) | 50 | 1 | 0.64 | 1.14 | 1.27 |
| Mimi (Défossez et al., 2024) | 50 | 4 | 0.85 | 1.64 | 3.03 |
| WavTokenizer (Ji et al., 2025) | 40 | 1 | 0.85 | 1.62 | 3.57 |
| X-codec2 (Ye et al., 2025) | 50 | 1 | 0.92 | 2.43 | 4.13 |
| BiCodec (Wang et al., 2025) | 50 | 1 | 0.92 | 2.51 | 4.18 |
| VIBEVOICE (Coupled) | 7.5 | N/A | 0.72 | 1.92 | 3.3 |
| VIBEVOICE (Acoustic) | 7.5 | N/A | 0.75 | 2.98 | 4.19 |

Table 7: Reconstruction results on LibriSpeech test-clean set. $N_q$ denotes the number of quantizers; VIBEVOICE uses a single continuous $\sigma$-VAE. Higher PESQ, STOI, and UTMOS scores indicate better performance.

presents a breakdown of the inference cost in milliseconds (ms) per generated segment for different VIBEVOICE model sizes (1.5B and 7B LLM parameters) and varying numbers of DDPM diffusion steps (1 and 10) for the diffusion head. The measurements include the time taken by the Large Language Model (LLM), the Diffusion Head, the Acoustic Decoder, and the Semantic Encoder, along with Real-Time Factor (RTF).

| Model | Model Size | Diffusion Step | Inference Cost (ms) | | | | RTF |
|---|---|---|---|---|---|---|---|
| | | | LLM | Diffusion Head | Acoustic Decoder | Semantic Encoder | |
| MoonCast | 1.5B | N/A | | | N/A | | 1.43 |
| Higgs Audio V2 | 3B | N/A | | | N/A | | 0.72 |
| VIBEVOICE | 1.5B | 1 | 47.87 | 2.69 | 14.96 | 14.71 | 0.62 |
| | | 10 | 53.80 | 22.76 | 14.89 | 14.59 | 0.83 |
| | 7B | 1 | 58.71 | 3.25 | 14.75 | 14.52 | 0.70 |
| | | 10 | 71.73 | 28.87 | 14.80 | 14.52 | 0.97 |

Table 8: Inference cost (ms) and Real-Time Factor (RTF) comparison between VIBEVOICE and baseline models. Measurements were conducted on a single NVIDIA A6000 GPU with a batch size of 1.

As observed from Table 8, the LLM component constitutes the most significant portion of the inference time, which is expected given its size and autoregressive nature. For the 1.5B model, the LLM takes approximately 48-54 ms, while for the larger 7B model, this increases to around 59-72 ms. The inference time of the Diffusion Head scales linearly with the number of DDPM steps; increasing the steps from 1 to 10 results in a roughly 8-9x increase in its processing time. The costs for the Acoustic Decoder and Semantic Encoder remain consistently low and efficient at around 14-15 ms.

Crucially, the Real-Time Factor (RTF) for VIBEVOICE remains consistently below 1.0 across all tested configurations, ranging from 0.62 to 0.97, signifying faster-than-real-time capability. In contrast, the baseline MoonCast exhibits an RTF of 1.43, indicating slower-than-real-time processing. While Higgs Audio V2 achieves a competitive RTF of 0.72, it relies on a parallel generation strategy (processing 3-second audio chunks concurrently). VIBEVOICE achieves

comparable or superior efficiency within a strictly streaming framework, making it well-suited for real-time applications and the generation of long-form audio without accumulating latency.

## F  TRAINING HYPER-PARAMETERS

Table 9 summarizes hyperparameters used for training the Tokenizer (both Acoustic and Semantic variants) and the main VIBEVOICE model.

| Stage | Hyper-parameters | Values |
|---|---|---|
| Tokenizer | Stages | 7 |
| | Blocks per stage | [3, 3, 3, 3, 3, 3, 8] |
| | Downsampling ratios | [2, 2, 4, 5, 5, 8] |
| | Adam $\beta$ | (0.8, 0.99) |
| | Adam $\epsilon$ | 1e-6 |
| | LR | $3 \times 10^{-4}$ |
| | Weight decay | 0.01 |
| | Gradient norm clip | 1000 |
| | Training steps | 300,000 |
| | *Acoustic Tokenizer* | |
| | VAE dim | 64 |
| | VAE $C_\sigma$ | 0.5 |
| | Batch size | 160s |
| | Sample size | 4s |
| | *Semantic Tokenizer* | |
| | Batch size | 1024 tokens |
| VIBEVOICE | Diffusion head layers | 4 |
| | Diffusion head FFN ratio | 3 |
| | Diffusion beta schedule | Cosine |
| | Diffusion train steps | 1000 |
| | Diffusion loss weight | 5 |
| | Adam $\beta$ | (0.9, 0.95) |
| | Adam $\epsilon$ | 1e-8 |
| | Batch size | 4M |
| | Learning rate | $1 \times 10^{-4}$ |
| | Learning schedule | Cosine |
| | Warmup steps | 500 |
| | Weight decay | 0.1 |
| | Gradient norm clip | 2 |
| | Training steps | 110,000 |

Table 9: Summary of training hyperparameters for Tokenizer and VIBEVOICE model stages.

## G  VIBEVOICE-OBJECTIVE EVALUATION

To rigorously evaluate the capabilities of VIBEVOICE in generating long-form, multi-speaker podcasts, particularly its scalability with varying speaker counts and durations, we curated a dedicated evaluation set named **VIBEVOICE-Eval**. This dataset was constructed using our internal data processing pipeline (as described in Section 2.3 of the main text) from podcast sources not included in the VIBEVOICE training set, ensuring a fair assessment of generalization.

Table 10 details the distribution of the VIBEVOICE-Eval dataset. It comprises a total of 108 distinct podcast segments, amounting to approximately 28.9 hours of audio. The dataset is stratified by the number of unique speakers present in each segment, ranging from 1 to 4 speakers. As shown, the average duration of segments tends to increase with the number of speakers, with 4-speaker segments

| Speaker Numbers | Samples Count | Average Duration (s) | Total Duration (h) |
|:---:|:---:|:---:|:---:|
| 1 | 30 | 878 | 7.32 |
| 2 | 35 | 905 | 8.8 |
| 3 | 27 | 984 | 7.38 |
| 4 | 16 | 1210 | 5.38 |
| Total | 108 | 962 | 28.9 |

Table 10: Distribution of the VIBEVOICE-Eval dataset, stratified by the number of speakers per sample. Durations are provided in seconds (average) and hours (total).

averaging over 20 minutes (1210.81 seconds). This diverse composition allows for a comprehensive evaluation of model performance across different levels of conversational complexity and length.

To provide a qualitative illustration of the content within the VIBEVOICE-Eval dataset, below is an excerpt from a sample segment featuring 3 speakers. This example showcases typical conversational turn-taking and dialogue flow present in the evaluation materials:

---

**Dialogue Example**

Speaker 0: Oh, look. There's Hank and Trash Truck.
Speaker 1: Hi, Walter.
Speaker 2: Hey, Hank. Hey, Trash Truck.
Speaker 1: Hi, Donnie.
Speaker 0: Look at me, Hank. I'm swinging. And you guys thought this rope wasn't strong enough to hold a bear. Mm-hmm. Sure, you can have a turn. I bet it's even strong enough to hold a trash truck.
Speaker 2: Nuh-uh, trash truck. I'm next. Now, come on, Walter, catch up. Oh, it's my turn. Yes!
Speaker 0: Oh. Okay.
Speaker 2: Yes, okay, I'm ready. Now shove me. Somebody shove me. I want to be shoved. Higher! Higher! Woohoo!
Speaker 2: Oh, I can see the top of Trash Truck's head from up here! Ah, ha, ha! Yes! Woo!
Speaker 1: Okay, Donnie. I think it's Trash Truck's turn now.
Speaker 2: No, no, no. It's still my turn. Yes.
Speaker 1: Well, at school, each person gets five pushes for a turn. Then, it's the next person's turn.
Speaker 2: What's a five?
Speaker 0: I don't know. Sounds complicated.
Speaker 1: It's a number. Every time you're pushed on the swing, we count a number. Walter, you keep pushing and Trash Truck and I will count.
Speaker 2: Trash Truck can count?
Speaker 1: Yeah, Trash Truck can count. Show him, Trash Truck.
Speaker 2: Wow, trash truck sure knows his numbers. I mean, hawks. Wow.
Speaker 1: Okay, your turn's up, Donnie.
Speaker 2: Oh, alright. Okay.
Speaker 0: Oh, okay. Who's next?
Speaker 2: Get on up there, trash truck. Come on, buddy.
Speaker 0: See? I told you, the rope is strong.

---

## H  SUBJECTIVE EVALUATION

The annotation interface is designed as the following:

We conducted subjective evaluations engaging multiple human listeners to assess the quality of generated podcasts. Listeners rated samples on a 5-point score across three key dimensions: Realism, Richness, and Preference. We used two popular and competitive open-source model: SesameAILabs-CSM (SesameAILabs, 2025) and Higgs Audio V2 (Boson AI, 2025), and two proprietary dialogue models: Elevenlabs v3 alpha (Elevenlabs) and Gemini 2.5 pro preview tts (Google).

Below are the definitions and detailed scoring standards for the six dimensions used in our subjective evaluation:

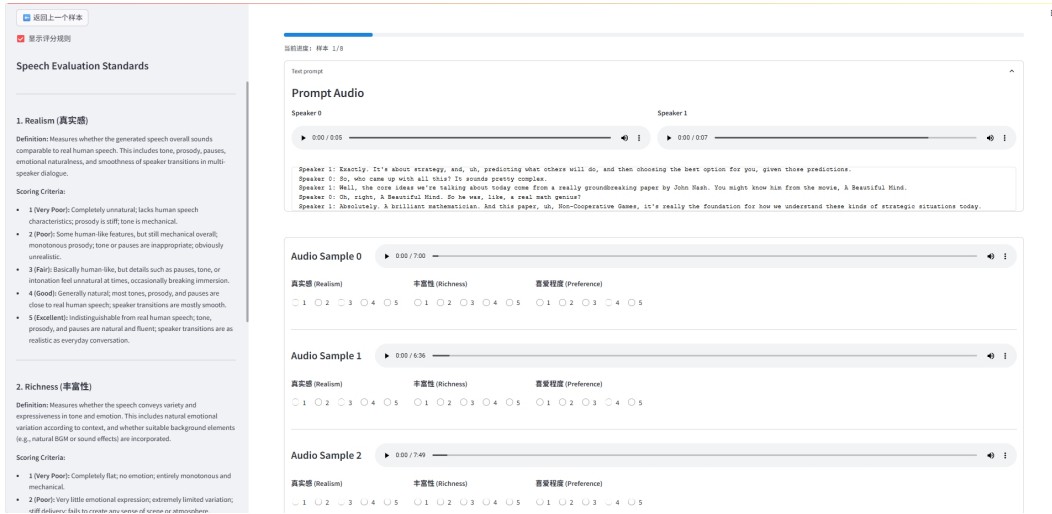

Figure 5: Annotation interface of subjective evaluation.

---

**Realism**

**Definition:** Measures whether the speech sounds naturally generated, rather than mechanical or overly deliberate. A high score indicates that the speaking style is closer to the natural rhythm, pauses, intonation, etc., of real human communication.

**Scoring Standard:**

- **1 point (Very Poor):** Speech is completely unnatural, sounding very mechanical, stiff, or robotic.
- **2 points (Poor):** Speech is noticeably unnatural, with clear problems in rhythm, pauses, intonation, or tone, sounding quite awkward.
- **3 points (Average):** Speech naturalness is acceptable, with occasional unnatural elements (e.g., improper pauses, flat intonation), but generally acceptable.
- **4 points (Good):** Speech is quite natural and fluent, close to human speech, but may have minor shortcomings in some details.
- **5 points (Very Good):** Speech sounds very natural and fluent, as if a real human is speaking, with excellent naturalness.

---

**Richness**

**Definition:** Measures whether the speech conveys variety and expressiveness in tone and emotion. This includes natural emotional variation according to context, and whether suitable background elements (e.g., natural BGM or sound effects) are incorporated.

**Scoring Standard:**

- **1 point (Very Poor):** Completely flat and monotonous; no emotion or variation; mechanical delivery.
- **2 points (Poor):** Very limited emotional expression; stiff tone; fails to create a sense of atmosphere or scene.
- **3 points (Average):** Some variation in tone or emotion, but insufficient to convey natural expressiveness; weak emotional coloring.
- **4 points (Good):** Emotionally expressive and relatively natural; noticeable changes in tone and intonation; conveys basic emotions and atmosphere.

> - **5 points (Very Good):** Emotionally rich and natural; tone, intonation, and emotional variation match the context; may include well-timed natural sound effects or background elements, creating strong immersion.

---

**Preference**

**Definition:** Measures the listener's overall subjective preference toward the speech. It reflects naturalness, pleasantness, attractiveness, and listenability—not just objective clarity or similarity.

**Scoring Standard:**
- **1 point (Very Poor):** Strong dislike; voice is grating, mechanical, or unpleasant; almost unbearable to continue listening.
- **2 points (Poor):** Dislike; noticeable problems such as stiffness, monotony, or unnaturalness; poor listening experience.
- **3 points (Average):** Neutral; acceptable but not engaging; no strong preference or dislike.
- **4 points (Good):** Like; natural and fluent; comfortable to listen to; provides some enjoyment.
- **5 points (Very Good):** Strongly like; highly attractive and enjoyable; pleasant enough to listen to for a long time.

## I   FIRST-PHASE EVALUATION RESULTS

| Model | Subjective | | | | | | Objective | |
|---|---|---|---|---|---|---|---|---|
| | Spontaneity | Coherence | Intelligibility | Quality | Similarity | Realism | WER | SIM-O |
| Cosyvoice2 (Du et al., 2024) | $3.15_{\pm1.06}$ | $3.09_{\pm1.21}$ | $3.83_{\pm1.02}$ | $3.17_{\pm1.02}$ | $3.15_{\pm1.11}$ | $3.10_{\pm1.03}$ | 3.45 | **0.68** |
| MoonCast (Ju et al., 2025) | $3.17_{\pm1.06}$ | $3.67_{\pm1.01}$ | $4.00_{\pm0.91}$ | $3.04_{\pm0.9}$ | $3.19_{\pm1.14}$ | $3.02_{\pm1.12}$ | 2.81 | 0.56 |
| **VIBEVOICE-1.5B** | $\textbf{3.86}_{\pm0.92}$ | $\textbf{3.89}_{\pm0.94}$ | $\textbf{4.40}_{\pm0.66}$ | $\textbf{3.97}_{\pm0.94}$ | $\textbf{3.66}_{\pm1.05}$ | $\textbf{3.78}_{\pm0.95}$ | **1.11** | 0.55 |

Table 11: Subjective and objective evaluation on podcast generation. For all subjective metrics and SIM-O, higher scores indicate better performance. For WER, lower scores are better.

To verify the effectiveness of proposed VIBEVOICE framework, we conducted our first-phase experiment comparing CosyVoice2, MoonCast, and VIBEVOICE-1.5B through both subjective and objective evaluations.

For the subjective evaluation, 20 human evaluators were invited to assess the quality of the generated podcasts. Each evaluator rated samples on a 5-point scale across six key dimensions: Spontaneity, Coherence, Intelligibility, Quality, Similarity (to the voice prompt), and Realism.

For every text input, we generated eight podcasts with each of the three models, yielding an average audio length of 7.3 minutes per sample. In total, each evaluator listened to approximately **2.9** hours of audio—roughly equivalent to reviewing **1,000** short utterances of 10 seconds each.

From Table 11, we can observe that:

(1) VIBEVOICE-1.5B outperforms both CosyVoice2 and MoonCast across all subjective evaluation metrics as well as in WER.

(2) Compared with CosyVoice2, which can only generate relatively short utterances (<30s) and produces long podcasts by concatenating these clips, both MoonCast and VIBEVOICE (with end-to-end podcast generation) demonstrate significantly better coherence, highlighting the effectiveness of E2E podcast generation for maintaining contextual and prosodic consistency.

(3) Another notable finding is that although CosyVoice2 achieves a higher score than VIBEVOICE on the objective SIM-O metric, human evaluators consistently rated VIBEVOICE as having higher

similarity. One possible explanation is that SIM-O primarily captures timbre similarity but neglects prosodic features such as intonation, rhythm, and expressiveness. In natural conversations, speakers often exhibit subtle variations in voice due to emotional shifts or pitch dynamics. These variations are typically perceived by human listeners as consistent with the original speaker, but are penalized by SIM-O. **When SIM-O exceeds a certain threshold, it becomes less sensitive to these human-perceived consistencies, making it insufficient for evaluating speaker identity retention in long-form, multi-turn speech synthesis.**

## J SPECTRAL ANALYSIS OF DIFFUSION INFERENCE STEPS

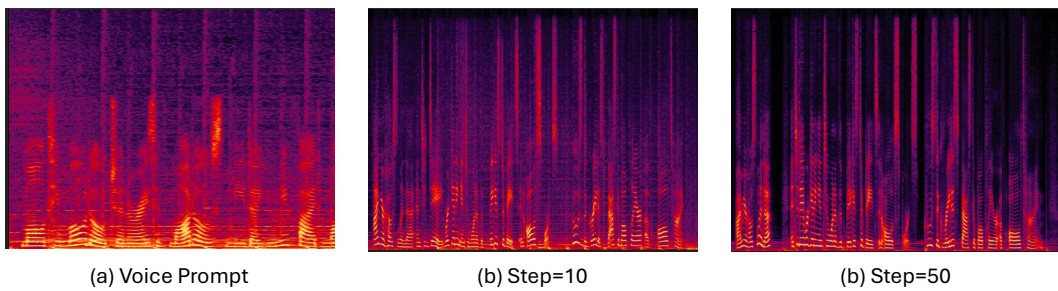

(a) Voice Prompt        (b) Step=10        (b) Step=50

Figure 6: Spectrogram analysis of the denoising process. (a) The **Voice Prompt** exhibits natural environmental textures (background noise, reverb). (c) At **Step=50**, the model aggressively strips these non-speech components, yielding a clean but spectrally divergent signal. (b) **Step=10** offers the optimal balance, preserving the environmental "atmosphere" necessary for high speaker similarity.

Diffusion-based decoding inherently acts as a progressive denoiser. While increasing inference steps ($T$) typically refines signal detail, it also biases the output toward an idealized, "clean" speech distribution. However, for "in-the-wild" podcast data, a speaker's identity is perceptually entangled with their acoustic environment (e.g., room reverberation, microphone response, and ambient noise floor).

Figure 6 visualizes how this denoising behavior affects acoustic consistency. The voice prompt (a) contains visible background spectral textures characteristic of natural recordings. At $T = 50$ (c), the model aggressively suppresses these non-speech components, producing audio that is technically cleaner (akin to a studio recording) but spectrally mismatched from the reference. This "over-cleaning" explains the counter-intuitive drop in SIM-O scores at higher steps: the metric penalizes the absence of the reference's environmental noise. In contrast, generation at $T = 10$ (b) retains these stochastic elements, maintaining the acoustic "atmosphere" of the prompt and thus achieving higher similarity scores.

## LLM USAGE STATEMENT

A large language model (LLM), namely *Gemini 2.5 Pro*, was employed solely for **grammar checking** of the manuscript. It was also used to generate dialogue text for the subjective evaluation of VIBEVOICE.

The LLM did **NOT** contribute to the research design, training data collection, analysis, or interpretation of results. The authors bear full responsibility for the accuracy and integrity of all content presented in this paper.

