# OpenReview forum: "VibeVoice: Expressive Podcast Generation with Next-Token Diffusion"
_ICLR.cc/2026/Conference — ICLR 2026 Oral_

### Official Review · Reviewer_jDkA · 2025-10-25

**Soundness:** 2
**Presentation:** 2
**Contribution:** 2
**Rating:** 2
**Confidence:** 5

**Summary:**

The paper introduces VibeVoice, a novel zero-shot Text-to-Speech (TTS) model designed for synthesizing long-form, multi-speaker conversational audio (e.g., podcasts, up to 30 minutes with up to 4 speakers). This addresses the limits of traditional TTS in handling scalability, speaker consistency, and natural turn-taking.

A core technical element is an ultra-low frame rate (7.5 Hz) continuous speech tokenizer that significantly boosts computational efficiency for long sequences while preserving audio fidelity. Using an extensive new dataset with pseudo transcriptions and turn-taking labels, VibeVoice employs a next-token diffusion framework. This enables the model to achieve a high degree of naturalness in turn-taking, pacing, and subtle non-lexical cue rendition (like breaths), crucial for authentic, immersive conversational output.

**Strengths:**

The paper addresses an important problem in speech generation. Synthesizing long-form, multi-speaker conversational audio is highly relevant for real-world applications such as podcast generation, enhanced virtual assistants, and dialogue systems. The scenario is undoubtedly innovative and highly applicable

**Weaknesses:**

The core weaknesses of this work lie in a lack of sufficient technical novelty and inconsistent experimental evaluation.

1. Methodology: The next-token diffusion (or next-distribution) framework, a central component of VibeVoice, has been previously introduced in the literature [1]. The paper does not clearly articulate the incremental technical novelty VibeVoice contributes beyond adapting this existing framework.

2. Scenario: The goal of synthesizing speech for multi-speaker (more than two people) scenarios, particularly for long conversational contexts, is also not entirely new. Prior work [3,4], has specifically designed and discussed methods for long conversational speech generation for podcasts and similar applications for more than 2 people [2].

3. While the paper addresses long-context dialogue generation, it fails to include a fair and direct comparison against key existing models that tackle similar challenges in multi-speaker and long-form synthesis [3,4]. This makes it difficult to ascertain the true performance gain of VibeVoice.

4. The 7.5Hz tokenizer appears to be the core technical contribution, yet no sufficient details are provided to fully evaluate its novelty or effectiveness

5. There is very little detail about the data preparation pipeline.

Moreover, there are many inconsistent and incomplete experimental evaluation:

1. Baseline Inconsistency: The choice of baselines across the evaluation tables is inconsistent. The paper should use a uniform set of relevant baselines across all comparative tables (e.g., Table 1 and Table 2) to provide a clear and comprehensive performance picture.

2. Missing Subjective Metrics: Table 1, which appears to focus on multi-speaker/turn-taking quality, lacks subjective (human) evaluation metrics for some systems.

3. Missing Single-Speaker Results: The evaluation in Table 2 is incomplete as it lacks results for the single-speaker scenario.

Finally, the paper fails to include a complete and up-to-date list of references for all cited or discussed works. This is an essential requirement for a conference paper.

[1] Zhu, Xinfa, Wenjie Tian, and Lei Xie. "Autoregressive speech synthesis with next-distribution prediction." arXiv preprint arXiv:2412.16846 (2024).

[2] Xie, Kun, et al. "Fireredtts-2: Towards long conversational speech generation for podcast and chatbot." arXiv preprint arXiv:2509.02020 (2025).

[3]Zeghidour, Neil, et al. "Streaming sequence-to-sequence learning with delayed streams modeling." arXiv preprint arXiv:2509.08753 (2025).

[4] Ju, Zeqian, et al. "MoonCast: High-quality zero-shot podcast generation." arXiv preprint arXiv:2503.14345 (2025).

[5] Zhang, Leying, et al. "CoVoMix: Advancing zero-shot speech generation for human-like multi-talker conversations." Advances in Neural Information Processing Systems 37 (2024): 100291-100317.

[6] Mitsui, Kentaro, Yukiya Hono, and Kei Sawada. "Towards human-like spoken dialogue generation between ai agents from written dialogue." arXiv preprint arXiv:2310.01088 (2023).

**Questions:**

1. VibeVoice claims to synthesize long-form audio with up to four speakers. Given the high risk of speaker timbre drift and identity confusion in long-context, zero-shot TTS, what explicit architectural mechanisms are implemented to ensure robust and consistent speaker similarity for all four voices?

2. How to prepare and collect data for more than two speakers? For transcription, it is unclear whether to rely only on pseudo-transcription, or should real data also be used? Furthermore, why doesn't pseudo-transcription negatively impact model performance?

---

> ### Author Response · Authors · 2025-11-20
>
> *[Q1]: Clarification on technical novelty beyond the existing framework [1].*
>
> [Answer] Thanks for the comment. We appreciate the reviewer’s perspective. However, we would like to clarify that [1] aligns more closely with the GIVT framework [7], and its methodology and objectives differ from those of VibeVoice:
> 1. The mentioned work[1] directly predict the next distribution (mean and variance), but VibeVoice integrates diffusion into LLM to denoise the next specific acoustic token, which tends to offer more powerful modeling expressivity.
> 2. The referenced work [1] is primarily validated on conventional short-utterance TTS, whereas VibeVoice is designed for a more realistic and challenging setting involving multi-speaker and long-form speech generation. Recent open-source systems such as FireredTTS2 [2], CovoMix [5], and MoonCast [3] still face challenges in long-duration generation, typically supporting a maximum length shorter than 12 minutes. In contrast, VibeVoice maintains stable performance over extended durations (up to 90 minutes) and achieves dialogue quality comparable to, or even surpassing, top-tier proprietary models such as Gemini 2.5 Pro (Table 1).
> 3. The mentioned work[1] proposed a flow constrained VAE, achieves PESQ 2.01 and UTMOS 3.66 at frame rate 12.5. Our trained acoustic tokenizer is more compact and effective under the same reconstruction benchmark, reaches PESQ 2.98 and UTMOS 4.19 at frame rate 7. Besides, In our revised manuscript (Appendix B, Table 8), we systematically compare Acoustic-only, Coupled, and Hybrid tokenizers. This analysis clarifies the technical roadmap for the community.
> 4. VibeVoice also introduces a transparent, automated pipeline (detailed in Appendix A) covering segmentation, diarization, and quality filtering. This contributes a reproducible methodology for constructing high-quality long-form datasets, a critical component often omitted or kept proprietary in prior literature and top-tier proprietary systems. In the revised version of our paper, we also added a comparison between our data pipeline against other widely-used data pipelines. (Appendix A, Table 5).
>
> [7] Tschannen, Michael, Cian Eastwood, and Fabian Mentzer. "Givt: Generative infinite-vocabulary transformers." ECCV, 2024.
>
> ---
>
> *[Q2] long-form multi-speaker scenario is not new, the absence of comparisons against specific works[2,3,4]*
>
> Thanks for the comment.
>  We would like to clearly clarify that the paper already **includes** comparisons against both recent open-source and commercial long-form systems, including **MoonCast [4]**, Higgs Audio V2, Gemini 2.5 Pro (Preview), and 11Labs V3 Alpha, as reported in Table 1.
>
> In particular, MoonCast [4] serves as our primary baseline. We conduct direct comparisons in both Table 1 and Table 2, and the results show that VibeVoice significantly outperforms MoonCast across multiple metrics.
> **Therefore, the claim that our work does not compare to prior long-form multi-speaker models may stem from a misunderstanding.**
>
> Regarding [2] and [3], these are concurrent works that appeared on arXiv in September 2025, the same month as the ICLR submission deadline for this paper. Even so, [2] primarily focuses on short conversational segments (around 3 minutes), whereas VibeVoice is designed for ultra-long dialogue generation (up to 90 minutes). The technical challenges of maintaining stability and coherence at these two scales are fundamentally different.
>
> ---
>
> *[Q3]: No sufficient details are provided to evaluate the effectiveness of the proposed 7.5Hz tokenizer*
>
> [Answer]  We respectfully point out that the effectiveness of the 7.5Hz tokenizer are comprehensively documented in the paper. We provide extensive objective evaluations on both LibriTTS (Table 3) and LibriSpeech (Appendix C, Table 8) against other widely used speech tokenizers.
>
> ---
>
> *[Q6] Missing Single-Speaker Results*
>
> [Answer] We respectfully point out a **factual oversight**. Table 2 explicitly includes a dedicated section for the single-speaker scenario (Columns 3-4, labeled "1 Speaker").
>
> ---
>
> *[Q7] Missing some references*
>
> [Answer] We respectfully clarify that the majority of discussed works were already present in the original bibliography. We have double-checked the manuscript and added any remaining citations to ensure completeness in the revised version.

---

> > ### Comment · Reviewer_jDkA · 2025-11-20
> > **Further Questions**
> >
> > I appreciate the authors' response and the revised manuscript.  Regarding the comparison with [1] and the tokenizer performance, I agree with the author's points. However, I still have several concerns, and I will clarify my questions to avoid confusion.
> >
> > 1. I believe the multi-speaker scenario is reasonable and novel, but the author's experimental comparison is unreasonable. The author conducted comparisons in Table 2, but except for the 2-speaker case, other scenarios don't even have baseline results, which means Table 2 has no baselines for comparison (except for CosyVoice). Since multiple models were compared in Table 1, I strongly suggest using one or more of those models for comparison in this scenario, rather than leaving it blank. Moreover, since the baseline system is Mooncast, why are the Mooncast results in Table 1 left blank? Is this intentional or are there other reasons?
> >
> > 2. Regarding the tokenizer, what I'm actually curious about is whether using a different tokenizer (such as DAC) instead of the 7.5Hz tokenizer, while keeping the same model architecture, could achieve better performance? This would help understand whether the gain comes from the model architecture or from the tokenizer. If the 7.5Hz tokenizer is not used, can the system still support such long speech generation? Does the long speech generation capability stem from the model itself or from the tokenizer?
> >
> > 3. I still strongly recommend that the author compare with FireRedTTS2. After all, this is a reasonable multi-speaker baseline system, and currently your paper does not include any baseline systems beyond the 2-speaker case. This is not mandatory, but I hope to see this type of experiment in the future.
> >
> > 4. I still think the author's explanation of the data pipeline still lacks details. I suggest the author could make their data processing pipeline public (though this is not mandatory). Btw, Appendix G is about evaluation and not about the author's data generation/processing. Please correct me if I misunderstood.
> >
> > 5. Regarding speaker drift, my concern is whether the speaker identity at the 1st minute and the speaker identity at the 60th minute are still the same person. Could some comparisons be conducted to verify this?
> >
> > In summary, while I acknowledge the innovative aspects of this work and the authors' efforts in the revision, I believe the experiments would benefit from more thorough baseline comparisons and clearer ablation studies to isolate the contributions of different components. I remain optimistic about this work and hope to see these issues addressed in the revision.

---

> ### Author Response · Authors · 2025-11-20
>
> *[Q8] Explicit architectural mechanisms used to prevent speaker timbre drift and ensure identity consistency in long-form generation.*
>
> [Answer] We respectfully highlight that **the entire VibeVoice framework is explicitly architected to solve this exact challenge**. As comprehensively detailed in Sections 2 and 3, we employ a synergistic approach rather than a single mechanism:
> 1. 7.5 Hz Ultra-low Frame Rate: This is the cornerstone for long-context consistency. It allows the LLM to fit the entire conversation history into its context window, ensuring it can "remember" and maintain speaker identity over time.
> 2. Hybrid Decoupled Representation: By separating semantic and acoustic streams, we prevent the "content-timbre entanglement" that typically causes identity confusion in single-stream models.
> 3. Next-Token Diffusion & 7B Scaling: The diffusion head ensures high-fidelity acoustic generation, while scaling to 7B parameters provides the necessary capacity to model complex Scenarios.
> 4. Data Pipeline: Our specialized pipeline (Section 2.3) provides the model with high-quality, diarized turn-taking examples to learn robust transitions.
>
> ---
>
> *[Q9] How to prepare and collect data for more than two speakers? For transcription, it is unclear whether to rely only on pseudo-transcription, or should real data also be used? Furthermore, why doesn't pseudo-transcription negatively impact model performance?*
>
> [Answer]  Thanks for the question.
>
> Our data preparation pipeline is detailed in Appendix A. For each speaker data, we run diarization to assign a speaker label to each segment; the diarization system can in principle handle 1–20 speakers per recording. However, in practice we observe that diarization accuracy degrades as the number of speakers grows. To ensure label reliability, we therefore only retain recordings with up to 4 speakers for training VibeVoice.
>
> Regarding transcriptions, all text used to train VibeVoice is pseudo-transcription: we generate transcripts with Whisper-large-v3-turbo and then apply additional filtering using a NeMo-ASR model to remove low-confidence or inconsistent segments. We do not mix in manually annotated transcripts in this work.
>
>
> Actually, using pseudo-labeled speech is now a common practice in large-scale speech and TTS systems; widely used corpora such as WenetSpeech [1], Emilia [2], and Libriheavy [3] are predominantly based on pseudo-transcriptions, and many recent TTS systems [4–6] trained on these corpora report competitive or state-of-the-art performance.
>
> Previous experiments suggest that high-quality pseudo-transcriptions are already sufficient for large-scale training. With strong ASR models and subsequent filtering, the obtained transcripts approximate human-level quality for most clean, non-overlapping segments.
>
> Nonetheless, pseudo labels are not error-free, and we expect fully human-annotated corpora to further enhance performance. The main challenge is cost: annotating tens of thousands of hours of multi-speaker audio with high-quality speaker labels is extremely time- and labor-intensive, and no such opensource dataset currently exists at this scale.
>
> [1] Binbin Zhang et al. "Wenetspeech: A 10000+ hours multi-domain mandarin corpus for speech recognition", ICASSP 2022
> [2] Haorui He, et al. " Emilia: An extensive, multilingual, and diverse speech dataset for large-scale speech generation", IEEE SLT 2024
> [3] Wei Kang et al. "Libriheavy: A 50,000 hours ASR corpus with punctuation casing and context"，IEEE 2024
> [4] Yushen Chen et at. "F5-TTS: A Fairytaler that Fakes Fluent and Faithful Speech with Flow Matching", ACL 2025
> [5] Yuancheng Wang et al. "Maskgct: Zero-shot text-to-speech with masked generative codec transformer", ICLR 2025
> [6] Dongya Jia et al. "DiTAR: Diffusion Transformer Autoregressive Modeling for Speech Generation", arxiv 2502.03930

---

> ### Author Response · Authors · 2025-11-20
> **1st Round FeedBack [Q1-3]**
>
> We would like to thank the reviewer for the timely response and constructive suggestions. Below we provide further clarifications.
>
> [Q1] Regarding baselines in Table 2: the MoonCast paper **only reports evaluation results in the two-speaker setting (Mooncast Paper Section 4.1.3)**, and its open-source repository and demo page also only provide examples for two-speaker generation. To our understanding, MoonCast is primarily designed for two-speaker conversational generation. Therefore, we include MoonCast only in the two-speaker setting in Table 2.
>
> In addition, **as noted in the caption of Table 1 (last line) in our first submission**, our early subjective evaluation—conducted on the same set of 8 samples—shows that Mooncast and CosyVoice perform worse than VibeVoice-1.5B (Details in Appendix I, Table 12). However, this preliminary evaluation was carried out before VibeVoice-7B was fully trained. Therefore, the evaluations presented in Table 1 and Table 12 were conducted in different batches. Since absolute subjective scores can vary depending on the test samples in comparison, **the absolute MOS values across the two tables are not directly comparable**. Nevertheless, the trend consistently indicates that VibeVoice-1.5B already outperforms both Mooncast and CosyVoice in terms of all subjective evaluation dimension.
>
> **Table 12,  Appendix I**
> | **Model**      | **Spontaneity** | **Coherence** | **Intelligibility** | **Quality** | **Similarity** | **Realism** | **WER** | **SIM-O** |
> |----------------|----------------|---------------|----------------------|-------------|----------------|-------------|---------|-----------|
> | **CosyVoice2** | 3.15±1.06      | 3.09±1.21     | 3.83±1.02            | 3.17±1.02   | 3.15±1.11      | 3.10±1.03   | 3.45    | **0.68**  |
> | **MoonCast**   | 3.17±1.01      | 3.67±0.95     | 4.00±0.87            | 3.04±0.90   | 3.19±1.14      | 3.02±1.12   | 2.81    | 0.56      |
> | **VibePod-1.5B** | **3.86±0.92** | **3.89±0.94** | **4.40±0.66**        | **3.97±0.94** | **3.66±1.05** | **3.78±0.95** | **1.11** | 0.55      |
>
>
> So, for fair comparison, In Table 1, we only shows the objective evaluation results of Mooncast and CosyVoice and put the  early subjective evaluation results in appendix and mentioned in the caption of Table 1.
>
> Since subjective evaluation for long-form generation is highly resource-intensive (one model requires roughly one hour of annotated audio) and our early subjective evaluation shows a clear trend, we decided not to include MoonCast in subjective evaluations after negotiation with our evaluation team.
>
> ---
>
> [Q2] Regarding the tokenizer comparison, we would like to clarify that VibeVoice uses an autoregressive diffusion model to directly generate continuous VAE latents (Section 2.2.3). In contrast, DAC encodes audio into discrete tokens via a VQ-VAE, which is fundamentally incompatible with our next-token diffusion formulation. Therefore, simply substituting DAC—or any discrete tokenizer—while “keeping the same architecture” is not feasible without redesigning the entire generation framework.
>
> As shown in Table 3, our tokenizer achieves significantly lower token rate while offering better reconstruction performance compared to DAC. This advantage largely stems from producing continuous latent representations, which preserve more fine-grained acoustic information relative to discrete tokens. To effectively model these continuous tokens autoregressively, VibeVoice incorporates the next-token diffusion framework. These two components—the continuous tokenizer and diffusion-based generation—are tightly coupled and jointly contribute to long-form speech generation performance.
>
> ---
>
> [Q3] Comparison with FireredTTS2. We fully acknowledge its relevance. However, FireredTTS2 supports generation of roughly three minutes of audio, whereas VibeVoice is designed for ultra-long dialogue generation up to 90 minutes. Direct comparison is therefore non-trivial due to the drastically different temporal scales. We agree this is a meaningful baseline for short multi-speaker settings, but re-design the test set is time consuming. We will consider including such comparisons in future extensions.
>
> ---

---

> > ### Comment · Reviewer_jDkA · 2025-11-23
> > **Reply to feedback**
> >
> > Thanks for the author's reply.
> >
> > I still have a concern with the experimental configuration, for example, the author still hasn't explained why Table 2 has no baselines for comparison (except for CosyVoice). Since multiple models were compared in Table 1, I strongly suggest using one or more of those models for comparison in Table 2, rather than leaving it blank. I don't think long evaluation time is a reason, and multiple tables can be confusing in the same paper.
> >
> > However, considering the author's responses on other aspects, I will slightly increase my score.

---

> ### Author Response · Authors · 2025-11-21
> **1st Round Feedback [Q4-5]**
>
> [Q4] Regarding the data pipeline, we apologize for the typo in our previous response. The detailed description of our data pipeline is provided in Appendix A. We are currently undergoing the internal permission process to open-source the data preprocessing pipeline, and we hope to release it soon once approval is granted.
>
> ---
>
> [Q5] On speaker drift:  We would like to clarify that Table 2 provides evidence that VibeVoice maintains speaker identity over long durations. Specifically, the speaker similarity metric (SIM-O) remains consistent between 0–12 minute and 12–30 minute segments. If speaker drift were occurring, we would expect a notable degradation in SIM-O in the latter segment, which is not observed.
>
> ----
>
> Thanks again for the timely response! Please feel free for further discussion.

---

> ### Author Response · Authors · 2025-12-01
>
> We sincerely thank the reviewer for raising the score after previous discussion.
>
> We would like to further clarify:
>
> Table 2 **does not** include only CosyVoice results. We do report MoonCast’s performance for the 2-speaker scenario. The reason for only providing the 2-speaker case initially is that the original MoonCast paper and released code focus exclusively on 2-speaker conversational speech generation, and do not describe or support multi-speaker cases.
>
>
> Following the reviewer’s suggestion, we have additionally evaluated MoonCast under 1, 3, and 4-speaker settings.
>
> **Word-error-rate WER**
>
> | Model              | Seq. | 1 Spk | 2 Spk | 3 Spk | 4 Spk | Overall |
> |--------------------|------|------:|------:|------:|------:|--------:|
> | MoonCast           | 40K  | 7.2   | 7.9   | 17.2‡ | 11.5‡ | 10.4‡   |
> | VIBEVOICE-1.5B     | 64K  | 0.63  | 1.92  | 1.48  | 1.34  | 1.22    |
> | VIBEVOICE-7B       | 32K  | **0.47** | **0.53** | **0.68** | **1.02** | **0.66** |
>
> ---
>
> **Speaker Similarity (SIM)**
>
> | Model              | Seq. | 1 Spk | 2 Spk | 3 Spk | 4 Spk | Overall |
> |--------------------|------|------:|------:|------:|------:|--------:|
> | MoonCast           | 40K  | 0.61  | 0.63  | 0.52‡ | 0.48‡ | 0.55‡   |
> | VIBEVOICE-1.5B     | 64K  | 0.63  | 0.59  | 0.58  | 0.58  | 0.60    |
> | VIBEVOICE-7B       | 32K  | **0.76** | **0.75** | **0.75** | **0.72** | **0.75** |
>
> ---
>
> As shown in the updated results:
> - MoonCast performs consistently between 1-speaker and 2-speaker cases
> - For 3-speaker and 4-speaker settings,  the performance of Mooncast model degrades significantly.
>
> We'd like to mention that MoonCast frequently crashes or produces invalid audio in 3–4-speaker generation. In many cases it outputs meaningless single-tone signals and fails to terminate. We evaluate MoonCast with up to three retries and only keep successful generations, which are the results reported in the table, labeled by **‡**.
>
> **We would also like to emphasize that the comparison for 3–4 speakers between VibeVoice and other models such as MoonCast is inherently unfair, since MoonCast was not designed for this scenario, which is also the case for other models shown in Table 1.**
>
> The main purpose of Table 2 is:
> - analyze ablation on tokenizer / model size / sequence length of proposed VibeVoice model
> - demonstrate the robustness and stability of VibeVoice under different lengths and speaker counts|
>
> We hope this clarification resolves the concern. Again, we truly appreciate the reviewer’s suggestions and feedback.

---

### Official Review · Reviewer_eyTk · 2025-10-25

**Soundness:** 4
**Presentation:** 4
**Contribution:** 4
**Rating:** 8
**Confidence:** 4

**Summary:**

VibeVoice is a next-token diffusion multispeaker long-context TTS model that benefit from the following innovations: 1) a 7.5 Hz framerate tokenizer; 2) an annotation pipeline that generates pseudo transcription and turn-taking labels.

**Strengths:**

1. this work studies an important problem - long-form conversation generation, and is one of the few papers in this field. The idea of using continuous features leads to significantly reduced sequence length which enables long context.
2. the data annotation pipeline is an important contribution

**Weaknesses:**

1. a big contribution of the the paper as noted in the abstract is data pipeline, but it's described in appendix, and there is not evaluation on the design choices of the data pipeline
2. the writing could be improved, for example, it should be noted that in the prompt part, the speech features are just short segments, while the text are the entire conversation (including both prompt and what's to be generated?)
3. missing RTF

**Questions:**

1. why do we use a $\sigma\text{-VAE}$ as opposed to just an Autoencoder for acoustic feature?
2. In appendix G, subjective eval interface is annotated with Chinese, are human listeners in the subjective evaluation native speakers of English?

---

> ### Author Response · Authors · 2025-11-20
>
> *[Q1] Details and evaluation of Data Pipeline.*
>
> [Answer] Thanks for pointing this out. We apologize for the limited visibility of the data processing pipeline in the main paper. Due to ICLR’s strict page limits and because the pipeline is a relatively independent component, we placed most implementation details in the appendix to maintain narrative coherence.
>
> Regarding evaluation, previous works on podcast generation[1,2,3] do not release implementation details or code of there data pipeline, and there is also a lack of publicly available benchmark datasets for long-form multi-speaker conversational podcast, making fair comparisons difficult.
>
> To address the corncern of data pipeline evaluation, we provide a comparative evaluation of our pipeline against two well-known audio data pipelines—WhisperX and Emilia—on three widely used public multi-speaker meeting datasets, reporting both diarization error rate (DER) and word error rate (WER). (For Emilia, we disabled the data-filtering step because it removes many audio samples.)
>
> **Diarization Error Rate (DER)**
>
> | Model      | AMI-IHM | AMI-SDM | AISHELL4 | AliMeeting | ALL   |
> |:-----------|:-------:|:-------:|:--------:|:----------:|:-----:|
> | WhisperX   | 18.27   | 23.05   | **14.55**| 35.53      | 22.00 |
> | Emilia     | 35.44   | 46.55   | 16.58    | 25.57      | 28.30 |
> | Ours       | **15.46**| **17.78**| 16.93   | **25.34**  | **18.84** |
>
>
> **Word Error Rate (WER)**
>
> | Model      | AMI-IHM | AMI-SDM | AISHELL4 | AliMeeting | ALL   |
> |:-----------|:-------:|:-------:|:--------:|:----------:|:-----:|
> | WhisperX   | 24.12   | 39.65   | 29.69| 36.62      | 32.90 |
> | Emilia     | 47.85   | 61.70   | 49.40    | 54.27      | 52.85 |
> | Ours       | **23.22**| **28.40**| **18.99**| **30.82**  | **25.43** |
>
> From the table we can observe that show that the proposed data pipeline consistently outperforms both baselines on both DER and WER on most of the datasets.
>
> However, we would like to clarify that:
> - Meeting datasets are intrinsically challenging (far-field microphones, low SNR, frequent speaker changes and overlaps), resulting in much higher DER/WER than normal podcast-style data. Actually, we avoid using any meeting data in the training of the VibeVoice model.
> -WER is sensitive to text normalization choices (e.g., punctuation, dates, URLs), so results should be interpreted cautiously. In this evaluation, we remove punctuation prior to scoring, following common ASR benchmarking practices (e.g., LibriSpeech, AMI, WenetSpeech).
> Therefore, these metrics mainly serve as partial indicators of pipeline robustness.
>
> In practice, we assess data quality through
> - Manual inspection, including random sampling and visualization of processed segments
> - Downstream TTS performance, where VibeVoice’s generation quality reflects the data pipeline reliability.
>
> [1] Ju, Zeqian, et al. "MoonCast: High-quality zero-shot podcast generation." arXiv preprint arXiv:2503.14345 (2025).
> [2] Zhang, Leying, et al. "CoVoMix2: Advancing Zero-Shot Dialogue Generation with Fully Non-Autoregressive Flow Matching", NeurIPS 2025.
>
> **This part has been added in the revised version of our paper.**
>
> ---
>
> *[Q2] Improving the clarity of the input*
>
> [Answer] Thank you for this valuable feedback. We have clarified this in our revision. To be precise, the acoustic prompt is a short audio segment used solely to provide the target voice and style (the "how"). The text input is the content to be synthesized (the "what"), and it **does not** include the transcript of the acoustic prompt.
>
> ---
>
> *[Q3] Missing RTF*
>
> [Answer]  Thank you for the suggestion. We have included the Real-Time Factor (RTF) analysis in **Table 9 (Appendix E) of the revised manuscript**.
>
>
> | Model      | Model Size | Diffusion Step | RTF |
> |:-----------|:-------:|:-------:|:--------:|
> | Mooncast  | 1.5B  | N/A   | 1.43|
> | Higgs Audio V2 | 3B     | N/A   | 0.72   |
> | VibeVoice       | 1.5B | 1| **0.62**|
> | VibeVoice       | 1.5B | 10| 0.83|
> | VibeVoice       | 7B | 1 | 0.70|
> | VibeVoice       | 7B | 10 | 0.97|
>
> Our RTF is computed using
>
> $$\text{RTF}=\frac{\text{Time Taken for Generation}}{\text{Duration of Audio Data}}$$
>
> The results show that VibeVoice achieves an RTF between 0.62 and 0.97 on a single NVIDIA A6000 GPU, which is competitive with recent state-of-the-art podcast generation models such as MoonCast and Higgs Audio V2, despite their similar or larger model sizes. This confirms that VibeVoice is capable of efficient, real-time streaming generation.
>
> ---

---

> ### Author Response · Authors · 2025-11-20
>
> *[Q4] Why do we use a sigma-vae as opposed to just an Autoencoder for acoustic feature?*
>
> [Answer] Thank you for this important question regarding our model design. We chose a $\sigma$-VAE over a deterministic autoencoder because our generative framework relies on modeling complex acoustic distributions, which benefits significantly from the **regularized, probabilistic latent space** that a VAE provides. However, standard VAEs are prone to "variance collapse" (where the posterior collapses to the prior) when paired with strong autoregressive models like LLMs. The $\sigma$-VAE mitigates this by enforcing a fixed variance, ensuring a stable and expressive latent space essential for high-fidelity synthesis.
>
> ---
>
> *[Q5] In appendix G, subjective eval interface is annotated with Chinese, are human listeners in the subjective evaluation native speakers of English?*
>
> [Answer] Thanks for your question. We would like to clarify that, as shown in Appendix G, **all evaluation instructions and criteria are annotated in both English and Chinese**.
>
> For the evaluations reported in our VibeVoice submission, the majority of human evaluators were native English speakers based in North America, with a minority being native Mandarin speakers who are fluent and professionally proficient in English.
>
> Actually, this evaluation interface is multilingual to supports both English and Mandarin listening tests, so you can find both Chinese and English annotation in the interface.

---

> > ### Comment · Reviewer_eyTk · 2025-11-22
> >
> > Thanks for your reply. I'm happy with most of the rebuttal, with only one concern regarding
> > > We chose a $\sigma$-VAE over a deterministic autoencoder because our generative framework relies on modeling complex acoustic distributions, which benefits significantly from the regularized, probabilistic latent space that a VAE provides.
> >
> > Could you show evidence for this statement?

---

> ### Author Response · Authors · 2025-11-26
>
> We are glad to hear that our previous responses have resolved most of your concerns. Regarding your remaining question on the evidence for the $\sigma$-VAE design choice, our explanation is as follows:
>
> 1. Why VAE instead of Deterministic AE?
>
> The core advantage of VAEs lies in the Evidence Lower Bound (ELBO) objective[1], which forces the model to learn a continuous data manifold rather than discrete points.
> *   AE Limitations: A deterministic AE minimizes reconstruction loss only ($\mathcal{L}_{AE} = ||x - D(E(x))||^2$). This often leads to a "broken" latent space where valid data points are separated by undefined regions ("holes"). If the dffusion predicts a latent vector that falls into these holes, the decoder produces severe artifacts.
> *   VAE Solution[1]: VAEs maximize the ELBO: $ \log p(x) \ge E_{q(z|x)}[\log p(x|z)] - D_{KL}[q(z|x) || p(z)] $
>     The KL-divergence term enforces the latent posterior $q(z|x)$ to align with a smooth prior $p(z)$. This regularization ensures Manifold Continuity[1,2], meaning the neighborhood of any valid latent vector is also valid.
>
> 2. Why $\sigma$-VAE instead of Standard VAE? (Robustness to Error Accumulation)
>
> While standard VAEs provide continuity, they often learn extremely small variances to maximize reconstruction fidelity. This becomes problematic in autoregressive settings due to error accumulation: During inference, the LLM and Diffusion head inevitably generate prediction errors that accumulate over long sequences.
>
> The $\sigma$-VAE enforces a larger fixed variance, effectively training the decoder to be robust to noisy inputs. We adopted this approach based on quantitative evidence from LatentLM [3]. The table below illustrates that appropriately increasing variance significantly improves class-to-image generation quality (lower FID is better) on ImageNet:
>
> | Latent Variance ($C_\sigma$) | Image Generation Quality (FID $\downarrow$) |
> | :--- | :--- |
> | $0.0008$ (Standard VAE Behavior) | $8.32$ |
> | $0.001$ | $6.43$ |
> | $0.01$ | $6.06$ |
> | $0.3$  | $5.31$ |
> | $1.0$ | $6.70$ |
> | $3.0$ | $8.19$ |
>
> The data reveals a clear trend. A standard VAE (mimicking variance $\approx 0.0008$) yields poor generation (FID $8.32$) because it cannot handle the prediction errors from the generative model. Guided by these findings, we empirically set $\sigma=0.5$ for VibeVoice (as noted in Table 10) to ensure stability in long-form synthesis.
>
> [1] Auto-Encoding Variational Bayes.
>
> [2] High-Resolution Image Synthesis with Latent Diffusion Models
>
> [3] Multimodal Latent Language Modeling with Next-Token Diffusion

---

### Official Review · Reviewer_Tqnw · 2025-10-27

**Soundness:** 4
**Presentation:** 3
**Contribution:** 4
**Rating:** 8
**Confidence:** 5

**Summary:**

This paper presents **VibeVoice**, a system designed for **long-form, multi-speaker conversational text-to-speech (TTS)** synthesis. The work addresses three major challenges in the field, **scalability**, **speaker consistency**, and **natural turn-taking with long-context stability**, particularly in the context of podcasts and other extended dialogues.

VibeVoice adopts a **hybrid acoustic–semantic representation**, where speech and text are encoded separately and later fused. The semantic stream is modeled using a fine-tuned **Qwen2.5 language model**, while the acoustic stream is generated via a **next-token diffusion model** that predicts continuous latent acoustic features at **7.5 Hz**.

In addition, the authors introduce a **customized data annotation pipeline** that includes segmentation, speaker diarization, and automatic filtering of low-quality samples.

Empirical results demonstrate improved **turn pacing**, **context-aware prosody**, and **speech naturalness** over long durations (up to 45 minutes), outperforming several leading commercial and open-source speech generation systems in both **subjective and objective evaluations**.

**Strengths:**

1. **Comprehensive System for Long-Form Conversational TTS**
- The paper presents a well-integrated end-to-end system capable of generating long, multi-speaker conversations with smooth and natural conversational flow, a significant advancement for practical TTS applications.

2. **Innovative Representation and Generation Mechanism**
- The use of a **7.5 Hz hybrid acoustic–semantic representation** effectively reduces context length for speech modeling.
- The **autoregressive next-token diffusion** architecture is a sound and well-motivated design choice for achieving high-quality, coherent speech synthesis.

3. **Robust Data Preparation Pipeline**
- The proposed data pipeline, combining segmentation, diarization, and transcript filtering, demonstrates careful system design and contributes to improving training data quality.

4. **Thorough and Convincing Evaluation**
- The model is extensively evaluated through both subjective and objective metrics.
- The **subjective evaluation includes 6 hours of audio** and **over 24 evaluators**, which makes the results convincing for an academic paper.
- The **blind subjective evaluation setup (Appendix G)** appears well-designed and appropriate for assessing perceptual quality.

**Weaknesses:**

Overall, I did not find any obvious weaknesses. There are minor typographical issues. For example, in **Table 3**, the last row appears to incorrectly include an *Nq* value even though the proposed speech tokenizer does not use a quantizer.

**Questions:**

1. Could the authors clarify the **key differences between VibeVoice and MoonCast**?
2. In the demo, the model appears capable of **singing**. Is this behavior intentional or emergent? Additionally, the singing quality seems suboptimal, are there potential ways to improve this aspect?

---

> ### Author Response · Authors · 2025-11-20
>
> *[Q1] Typo issues*
>
> [Answer] Thanks for your comment. We appreciate the reviewer pointing this out. We have updated Table 3 and replaced the value of $N_q$ in the last row with “–”. We will also review the manuscript carefully to correct other typographical issues.
>
> ---
>
> *[Q2] Differences between VibeVoice and MoonCast*
>
> [Answer] We sincerely thank the reviewer for raising this important point. MoonCast is indeed a notable prior work on multi-speaker dialogue audio generation. However, our work differs from MoonCast in several key aspects, summarized as follows.
>
> **1. Speech Tokenizer**
> VibeVoice uses a 64-dimensional continuous speech tokenizer with a frame rate of 7.5 Hz, while MoonCast adopts a discrete tokenizer with a codebook size of 8192 and a frame rate of 50 Hz.
>
> As a result:
>
> - The token sequence generated by VibeVoice is approximately 6.7× shorter, improving generation efficiency and enabling longer audio generation.
> - The tokens produced by VibeVoice have a significantly higher bitrate, allowing them to encode more fine-grained acoustic information.
>
> | Model    | Tokenizer Type | Frame Rate (Hz) | Bitrate (bits/sec) |
> |----------|----------------|-----------------|--------------------|
> | VibeVoice  | Continuous     | 7.5             | 7680               |
> | MoonCast | Discrete       | 50              | 650                |
>
> **2. Model Architecture**
>
> To support autoregressive generation using continuous speech tokens, VibeVoice adopts a next-token diffusion framework that integrates diffusion heads into an AR Transformer decoder.
>
> In contrast, MoonCast uses a standard autoregressive Transformer decoder to generate discrete speech tokens, similar to prior works such as CosyVoice2 and SeedTTS.
>
> **3.Generation Capability and Performance**
> ibeVoice can generate long-form multi-speaker audio involving 1 to 4 speakers for up to 30 minutes. MoonCast is limited to 2 speakers and a maximum length of roughly 10 minutes.
>
> Across both objective and subjective metrics, VibeVoice also demonstrates stronger voice quality, speaker consistency, and long-context stability.
>
> In summary, while MoonCast is a valuable prior work, VibeVoice advances both tokenization and modeling approaches to support longer, multi-speaker dialogue generation at higher audio quality.
>
> ---
>
> *Q3. Ways to improve sing ability*
>
> [Answer] Thanks for the question and for taking the time to listen to our demos.
>
> The model's singing ability is an emergent behavior—we did not design or explicitly model singing patterns. The model only receives plain text as input, and it spontaneously switches between speaking and singing based on the semantic content of the text. Importantly, we did not provide any explicit instruction or label regarding singing during either training or inference. This suggests that VibeVoice can produce coherent, spontaneous, and context-aware acoustic outputs beyond simple spoken speech.
>
> Regarding how to improve singing performance, we see two potential directions:
>
> - In current VibeVocie model, we did not deliberately include music or singing-oriented data in the training corpus, so the model has no specialized knowledge about how to sing well. Incorporating curated singing data could help improve melody control and vocal style quality.
>
> - In practice, we observe that the 7B model performs notably better than the 1.5B model in expressive settings such as emotional speech and singing. This suggests that further scaling may improve expressive capabilities, including singing.

---

> > ### Comment · Reviewer_Tqnw · 2025-11-24
> >
> > I have carefully reviewed the authors’ response.
> > My concerns regarding the novelty of VibeVoice relative to MoonCast have been addressed.
> > Based on the following strengths:
> > 1. VibeVoice demonstrates strong potential for unifying currently separate speech and music/sing generation models
> > 2. The proposed model is capable of generating coherent long-form speech of up to 90 minutes, which has achieve production level usage for podcast generation.
> > I suppose this paper is worthy of acceptance at ICLR 2026.
> > Accordingly, I have decided to raise my score.

---

> > > ### Author Response · Authors · 2025-11-26
> > >
> > > We sincerely thank the reviewer for the thoughtful reassessment and the decision to raise the score. We are greatly encouraged by your recognition of VibeVoice's strengths, particularly its potential to unify audio generation modalities and its proven capability for production-level long-form synthesis.
> > >
> > > As you noted, while proprietary models (e.g., OpenAI, Gemini) demonstrate impressive capabilities, their underlying techniques remain largely opaque "black boxes." **Our primary motivation with VibeVoice was to demystify these capabilities and provide the research community with a transparent, scalable, and reproducible framework for high-fidelity podcast generation**. We are thrilled that our technical contributions—from the 7.5Hz tokenizer to the robust data pipeline—have resonated with you, and we hope VibeVoice can serve as a solid foundation for future open research in long-context audio modeling.

---

### Official Review · Reviewer_WPLm · 2025-10-30

**Soundness:** 3
**Presentation:** 3
**Contribution:** 3
**Rating:** 8
**Confidence:** 4

**Summary:**

This paper proposes a method for generating long-form, multi-speaker audio podcasts. It introduces a continuous hybrid (semantic and acoustic) speech tokenizer that achieves high-quality audio reconstruction at a frame rate of 7.5 Hz. Building upon this, VibeVoice employs next-token diffusion to enable high-quality audio generation.

**Strengths:**

The authors design a continuous hybrid (semantic and acoustic) speech tokenizer that achieves high-quality audio reconstruction at 7.5 Hz.
They leverage next-token diffusion to enable high-quality audio generation.
An efficient pipeline for processing raw podcast data is established.
The method achieves impressive results compared to top-tier models.

**Weaknesses:**

The proposed method in this paper draws heavily on the design of LatentLLM. Moreover, the designs of both the semantic tokenizer and the acoustic tokenizer have already been extensively explored in numerous audio-related works.

**Questions:**

Why does the acoustic tokenizer perform well with a single speaker, but exhibit a significant drop in WER when a second speaker is introduced? Could this be due to insufficient training on multi-speaker data?

---

> ### Author Response · Authors · 2025-11-20
>
> We are very grateful for your strong support and recognition. We are glad that you recognized our pipeline's efficiency and the method's ability to achieve impressive results comparable to top-tier proprietary models.
>
> *[Q1] Differentiation from LatentLLM and justification for tokenizer design.*
>
> **RE:** We acknowledge that VibeVoice builds upon the foundational paradigm of LatentLLM. However, adapting this framework to long-form, multi-speaker podcast generation required solving challenges that the original LatentLLM did not address. Our contributions lie in:
> 1. Scaling and Setting: LatentLLM was a proof-of-concept. VibeVoice scales the architecture to 7B parameters for a significantly more realistic and challenging setting: 90-minute multi-speaker podcasts. While the recent open-source SOTA, MoonCast, struggles beyond 12 minutes, VibeVoice maintains stability throughout long durations (90 minutes) and achieves dialogue quality matching top-tier proprietary models like Gemini 2.5 Pro (Table 1). Also, we built a long-form audio processing pipeline for this goal.
> 2. Tokenizer Designs: While various tokenizer architectures exist, their efficacy in long-form multi-speaker generation remained unexplored. In our revised manuscript (Appendix B, Table 6), we systematically compare Acoustic-only, Coupled, and Hybrid tokenizers. This analysis clarifies the technical roadmap for the community.
>
> ---
>
> *[Q2] Why does the Acoustic tokenizer fail on 2+ speakers? Is it data insufficiency?*
>
> **RE:** We have empirically ruled out data insufficiency. As shown in our new ablation study (Appendix B, Table 6), all tokenizer variants were trained on the exact same dataset. The performance drop is strictly architectural:
> - Entanglement: The Acoustic-only tokenizer entangles speaker timbre with linguistic content. In multi-speaker scenarios, the decoder fails to disentangle "who is speaking" from "what is being said," leading to semantic collapse (high WER).
> - Validation via Coupled Tokenizer: We further validated this by testing a "Coupled" tokenizer. While it stabilized content, it failed to model diverse identities (SIM-O dropped to 0.45).
>
> Therefore, the failure is not due to a lack of data, but the necessity of the Hybrid architecture to explicitly decouple semantic planning from acoustic realization.

---

### Official Review · Reviewer_1JL1 · 2025-10-31

**Soundness:** 4
**Presentation:** 4
**Contribution:** 4
**Rating:** 8
**Confidence:** 4

**Summary:**

This paper introduces a new model VibeVoice  for long-form zero-shot text-to-speech.
VibeVoice can generate up to 90 minutes of speech from a text script and a voice-condition, supporting up to 4 speakers and outperforms SOTA models on podcast generation (as shown by quantitative and qualitative metrics).

This method relies on key design choices:
- a separation of acoustic and semantic tokens to get the best of both worlds between text intelligence and acoustic reconstruction
- ultra low frequency (7.5 Hz) tokens (this speeds up LLM inference time)
- next-token-diffusion objective (continuous targets) as opposed to the standard cross-entropy objective (discrete targets).

The authors show extensive ablatation experiments to validate their design choices and compare their generations to state of the art TTS models. They provide detailed training details, code and sample outputs (long-form generated conversations) for reproducibility.

**Strengths:**

- SOTA model quality on long-form podcast generation
- Great level of detail on all fronts : tokenizer training, LLM training, data preprocessing, inference settings
- original contribution of concatenating acoustic and semantic tokens for LLM input - with a soft target (diffusion). They do ablations on those design choices to validate them
- extensive ablation studies on design choices (tokenizer choice, base model size, diffusion inference settings)
- extensive evaluation on the model, with a new benchmark VIBEVOICE-Eval  + subjective human evaluation

**Weaknesses:**

- The WER metric on the CFG figure is not very stable, there are no clear trends (as opposed to SIM-O which shows clear trends). This questions the robustness of WER metric in the comparison figures to other models.
- This paper compares to other models in terms of performance but not speed, the  inference time appendix does not show other models as comparison

**Questions:**

clarifications:
- what is the diffusion training schedule and loss weighting that was used? Same question regarding inference schedule
- what number of diffusion steps was used for all metrics reported in comparison table?
- on the tokenizer ablation , it looks like using hybrid only shows benefits at 2+ speakers, while acoustic is better at 1 speaker. Could you elaborate on this?

---

> ### Author Response · Authors · 2025-11-20
>
> We are very grateful for your strong support and recognition. We particularly appreciate your recognition of our core design choices—specifically the 7.5 Hz ultra-low frame rate and the hybrid tokenization strategy—as key enablers for SOTA long-form podcast generation.
>
> ---
>
> *[Q1] The WER metric on the CFG figure lacks a clear trend compared to SIM-O, questioning the robustness of the comparison.*
>
> **RE:** The WER trend is not random but follows a consistent "U-shaped" trajectory centered around Step 10. This reflects the diffusion denoising dynamics:
> 1. Step < 10 (Under-denoised): At very few steps (e.g., 5), the output retains residual high-frequency noise. While semantically formed, these artifacts interfere with the ASR model's phoneme recognition, causing higher WER.
> 2. Step = 10 (Optimal): This is the "sweet spot" where linguistic content is fully resolved, and the noise floor is cleared without degrading the speech signal.
> 3. Step > 10 (Over-cleaned): As noted in our revision (Appendix J), excessive diffusion steps strip away natural paralinguistic textures and "roughness" typical of podcasts. This over-smoothing negatively impacts ASR segmentation, leading to a slight rebound in WER.
>
> ---
>
> *[Q2] Comparison of inference speed against other models.*
>
> **RE:** We have added Appendix E (Table 9) to provide a detailed breakdown of inference costs. The table below summarizes the Real-Time Factor (RTF) measured on a single NVIDIA A6000 GPU.
>
> | Model | Size | Diff. Steps | RTF ↓ |
> | :--- | :---: | :---: | :---: |
> | MoonCast | 1.5B | N/A | 1.43 |
> | Higgs Audio V2 | 3B | N/A | 0.72 |
> | VibeVoice | 1.5B | 10 | 0.83 |
> | VibeVoice | 7B | 10 | 0.97 |
>
> As shown, VibeVoice achieves real-time generation (RTF < 1.0) across both model sizes under default settings. VibeVoice achieves comparable or superior efficiency  within a streaming framework (even without using special optimazation), making it well-suited for the generation of long-form audio.
>
> ---
>
> *[Q3] Clarification on diffusion training/inference configurations.*
>
> **RE:**  We have updated Section 3.2 and Table 10 with these details:
> 1. Training: We utilized a Cosine beta schedule with a loss weight of 5.
> 2. Inference: VibeVoice were evaluated using 10 diffusion steps (with a CFG scale of 1.3) by default.
>
> ---
>
> *[Q4] Trade-off: Why is Acoustic-only better for 1-speaker, but Hybrid better for 2+?*
>
> **RE**  This performance shift reflects our architectural evolution path, which we have detailed in the new Appendix B:
> 1. Acoustic-only: In single-speaker settings, the entangled representation is efficient. However, in multi-speaker scenarios, it suffers from "semantic collapse" (WER spikes from 1.06 to 6.15) because the model fails to disentangle speaker identity from linguistic content.
> 2. Coupled (Intermediate attempt): We attempted a Coupled design (shared encoder for semantic+acoustic) to fix the semantic stability. While it improved WER, the shared compact latent space struggled to encode diverse identities, causing a severe drop in speaker similarity (SIM-O dropped to 0.45).
> 3. Hybrid (Final choice): The Hybrid architecture explicitly separates these streams. Although this separation adds slight complexity (marginally impacting the simplest 1-speaker case), it is the only configuration that prevents semantic collapse while maintaining high speaker fidelity in complex multi-speaker podcasts.

---

### Official Review · Reviewer_h5UP · 2025-11-01

**Soundness:** 3
**Presentation:** 3
**Contribution:** 3
**Rating:** 6
**Confidence:** 4

**Summary:**

The paper presents VibeVoice, a framework for zero-shot, expressive, long-form, multi-speaker podcast generation. It introduces two ultra-low frame rate latent (acoustic and semantic) operating at 7.5 Hz and integrates them into a large language model with a next-token diffusion mechanism for scalable, high-fidelity audio synthesis. The system is capable of generating podcasts up to 90 minutes with up to four speakers.

**Strengths:**

1. The paper tackles an important and challenging task—long-form podcast generation, which combines the difficulties of expressive multi-speaker speech synthesis, discourse coherence, and long-context modeling. Addressing such a complex problem is both timely and valuable for advancing speech generation research.
2. The proposed approach is methodologically reasonable and well-motivated. The reduction of frame rate to 7.5 Hz effectively shortens the sequence length for long-context modeling. The design choices—including (1) the use of continuous latent representations rather than discrete tokens, (2) the hybrid modeling of acoustic and semantic features, and (3) the introduction of a diffusion head to improve latent prediction—all contribute to handling the challenges of long-form expressive audio generation.
3. The evaluation includes a variety of subjective metrics.

**Weaknesses:**

1. Unclear Training Data Source and Scale: The training data is vaguely described as an internal pseudo-labeled podcast collection without specifying its source, composition, or duration. While the paper mentions ~80 billion training tokens, this could roughly correspond to around 3 million hours of audio, which is an enormous scale. Clarifying the dataset origin and scale is critical for assessing the reproducibility, fairness, and ethical validity of the work.

2. Lack of Analysis on Frame Rate (7.5 Hz) Choice. The paper repeatedly emphasizes the importance of the 7.5 Hz frame rate but does not justify why this particular rate is optimal. It would be highly informative to include experiments at alternative frame rates (e.g., 12.5 Hz or 15 Hz) to show how longer sequences affect LLM modeling difficulty, as well as lower rates (e.g., 3.75 Hz) to measure the trade-off between sequence compression and degradation in WER, expressiveness, or audio quality.

3. Limited Discussion on Scaling and Comparability. Since most compared baselines (e.g., ElevenLabs, Gemini) are black boxes with unknown data and architectures, the fairness of comparison is somewhat constrained. It would be helpful if the authors could supplement scaling experiments, such as: (1) Varying the amount of training data to show scaling behavior. (2) Increasing model capacity (e.g., from 7 B to 30 B) to explore potential performance gains. Such results would make the conclusions more robust and generalizable.

**Questions:**

1. I wonder to know that should we feed the text scripts at the beginning of sequence all at once (like VibeVoice), or in an interleaved manner (e.g., alternating segments of text and audio, as in FireRedTTS2)? Which one is better?

2. The paper frequently refers to “tokenizers,” but the representations used are actually continuous latent variables, not discrete tokens. It might be clearer to use terminology such as latent encoders or continuous representation extractors to avoid confusion.

3. In Figure 3(b), the SIM-o metric decreases as the number of diffusion steps increases. Why does this happen? Intuitively, more denoising steps might be expected to yield smoother or more consistent outputs—so the observed negative correlation deserves explanation.

**Details Of Ethics Concerns:**

The training data is vaguely described as an internal pseudo-labeled podcast collection without specifying its source, composition, or duration. While the paper mentions ~80 billion training tokens, this could roughly correspond to around 3 million hours of audio, which is an enormous scale. Clarifying the dataset origin and scale is critical for assessing the reproducibility, fairness, and ethical validity of the work.

---

> ### Author Response · Authors · 2025-11-20
>
> We sincerely thank the reviewer for recognizing the timeliness and value of our work in addressing the challenging task of long-form podcast generation.
>
> *[Q1] Training data details*
>
> **RE:** Thanks for the question. We clarify that the reported 80B training tokens include both text tokens and continuous speech tokens. The total duration of training audio is approximately 1.6M hours after preprocessing.
>
> Our data is mainly sourced from Emilia/Emilia-YODAS and LibriLight. We do not directly use their released segmented speech; instead, we reprocess the original raw recordings using our own pipeline. This results in ~0.45M hours from Emilia (vs. ~0.2M hours in their processed version, which performs more aggressive filtering) and ~0.06M hours from LibriLight, totaling ~0.51M hours.
>
> In addition, most recordings in the original datasets are shorter than 15 minutes. To support training for long-form multi-speaker dialogue generation, we construct long recordings by concatenating 2–4 short utterances while ensuring the number of distinct speakers ≤ 4. Each synthetic long recording is up to 90 minutes, and longer part is chunked. This expansion yields ~1.6M hours of training data.
>
> ---
>
> *[Q2] Tokenizer frame rate analysis.*
>
> **RE:** We evaluated tokenizers with frame rates of 15, 7.5, and 3.75. Both 15 and 7.5 achieved comparable performance in reconstruction quality and TTS, whereas 3.75 showed substantial degradation. Because the 7.5 frame-rate tokenizer can generate speech twice as long as the 15 frame-rate tokenizer for the same sequence length, we selected 7.5 as the most optimal configuration.
>
> | Frame Rate | Mel Dist. ↓ | PESQ ↑ | STOI ↑ | VISQOL ↑ | UTMOS ↑ | WER ↓ |
> | :--- | :--- | :--- | :--- | :--- | :--- | :--- |
> | 15 | 0.813 | 2.724 | 0.926 | 4.268 | 3.491 | 1.2 |
> | 7.5 | 0.798 | 2.756 | 0.929 | 4.289 | 3.505 | 1.2 |
> | 3.75 | 0.852 | 2.533 | 0.916 | 4.165 | 3.460 | 1.7 |
>
> (Columns 2–6 report reconstruction results; the last column shows TTS performance measured by WER.)
>
> ---
>
> *[Q3] Discussion on scaling and comparability.*
>
> **RE:** Due to GPU resource limitations, we did not conduct experiments with models larger than 7B parameters. Below are the results for models with 1.5B, 3B, and 7B parameters. We can observe that as the model size increases, there are significant improvements in WER and speaker similarity. Our subjective evaluation also indicates that the 7B model exhibits noticeably greater expressiveness. Similarly, because of GPU constraints, we did not perform extensive data analysis, we plan to explore this further to enhance the model in future work.
>
> | Model            | WER ↓     | SIM-O ↑     |
> | :--------------- | :-------------------: | :-------------------: |
> | VibeVoice-1.5B   | 2.11                 | 0.59                 |
> | VibeVoice-3B     | 1.21                 | 0.61                 |
> | VibeVoice-7B     | 0.66                 | 0.75                 |
>
> ---
>
> *[Q4] Feed all text upfront or interleave text and audio.*
>
> **RE:** If the text scripts are predetermined, an interleaved format is unnecessary. While interleaving is feasible, FireRedTTS2 demonstrated effectiveness for sequences of about 3 minutes, but its performance for longer durations remains uncertain. In contrast, VibeVoice supports hours-long speech generation and is also simpler from an engineering perspective.
>
> ---
>
> *[Q5] Terminology: "tokenizer" vs. "continuous latent variables".*
>
> **RE:** We agree that "tokenizer" traditionally implies discrete units. We used it to describe the functional role of segmenting audio for the LLM input. To prevent confusion, we have updated Section 2.1 to explicitly define this module as a "Continuous Tokenizer" that extracts continuous latent representations, distinguishing it from discrete vector quantization.
>
> ---
>
> *[Q6] Why does SIM-O decrease as diffusion steps increase?*
>
> **RE:**  We analyze this in Appendix J (Figure 6) of revised manuscript. Our eval data contains natural background noise. The diffusion head acts as a denoiser: higher steps (e.g., 50) aggressively clean the audio, removing the background textures present in the reference prompt. Lower steps (e.g., 10) retain more of these environmental characteristics. Since SIM-O measures spectral similarity (including noise profiles), the "cleaner" speech at high steps mathematically diverges from the noisy reference, resulting in lower scores despite higher intelligibility.

---

### Author Response · Authors · 2025-11-20
**From the Authors**

We would like to thank all reviewers and the Area Chairs for their time and insightful feedback throughout the review process.

Following the received comments, we have updated the manuscript with several substantial revisions (All revisions are highlighted in red), including:
 -  evaluation of the data pipeline (Appendix A, Table 5),
 -  additional analyses and experiments on the speech tokenizer design (Appendix B and D),
 -  the Real-Time-Factor (RTF) of VibeVoice (Appendix E, Table 9).

We have responded to the reviewers’ questions and comments, and we hope our clarifications can help address the raised concerns.

We welcome further discussion and appreciate the reviewers’ thoughtful engagement.

Thank you again for your time and consideration.

---

### Meta-Review · Area_Chair_Pfje · 2026-01-03

**Summary:**

The paper present a framework for long multi-speaker content generation. The framework is build from ground up, from the data to the tokenizer and generation model.

In general reviewers acknowledge importance of the problem (h5UP, Tqnw, eyTk, jDkA), substantial technical contribution (h5UP, WPLm, Tqnw, 1JL1), good evaluation  (h5UP, Tqnw, 1JL1), valuable data pipeline (WPLm, Tqnw, eyTk) and quality of the results (1JL1, WPLm). Almost all initial reviews was positive, authors also did a good job during the discussion phase by providing bunch of additional ablations (see response to h5UP).

One notable exception was reviewer jDkA, who initially was very critical of the paper, however later during the discussion phase increased the score.

Overall area chair believe that this is a good paper deserved to be accepted.

**Reviewer Concerns:**

The main set of concerns was raised by h5UP and jDkA.

h5UP:

**Unclear Training Data Source and Scale** - authors provide required info during dissuasion.

**Lack of Analysis on Frame Rate (7.5 Hz) Choice.** - authors provide additional evaluations.

**Limited Discussion on Scaling and Comparability** - authors provide additional evaluations.

Area chair believe most of the reviewers concerns were addressed.

jDkA:

**Novelty of Methodology, Scenario, Tokenizer** - authors provide a good summary of the differences with respect to prior works. Area chair acknowledge however, that novelty may be hard to judge properly. Thus area chair prefer not to include these concerns in the evaluation.

**Direct comparison with some baselines** - authors provide comparison with MoonCast, while note that Fireredtts-2 and Streaming sequence-to-sequence are concurrent works. Area chair found this response to be satisfactory.

**Some missing details and ablations** - authors provide additional details during the discussion and in supplementary, which area chair founds reasonable.

Area chair believe authors made a reasonable effort to address reviewer concerns.

**Reviewer Scores:**

h5UP - raise to 8

1JL1 - same

WPLm - same

Tqnw - same

eyTk - same

jDkA - raise to 4

---

### Decision · Program_Chairs · 2026-01-26

Accept (Oral)